# FOXK2 in skeletal muscle development: a new pathogenic gene for congenital myopathy with ptosis

Peixuan Wu[1,7], Nan Song[1,7], Yang Xiang[1,7], Zhe Tao[2], Bing Mao[3], Ruochen Guo[4], Xin Wang[1], Dan Wu[1], Zhenzhen Zhang[1], Xin Chen[1], Duan Ma [ID][1✉], Tianyu Zhang [ID][1✉], Bingtao Hao [ID][5,6✉] & Jing Ma [ID][1✉]

## Abstract

Congenital ptosis, a genetic disorder involving levator palpebrae muscle dysfunction, is often associated with congenital myopathy. The genetic causes of this condition remain poorly understood. In this study, we identified *FOXK2* mutations in five pedigrees with congenital myopathy and ptosis through whole exome sequencing and Sanger sequencing. Zebrafish with *foxk2* deficiency exhibited underdeveloped skeletal muscles and reduced mobility, while mice with *Foxk2* deletion in skeletal muscle stem cells (MuSCs) showed generalized skeletal muscle abnormalities. Further analysis revealed that FOXK2 deficiency impaired myogenic differentiation in C2C12 cells and disrupted mitochondrial homeostasis in both mouse MuSCs and C2C12 cells. Rescue experiments confirmed the loss-of-function effects of *FOXK2* mutation. Coenzyme Q10 treatment improved mitochondrial function and alleviated skeletal muscle development defects in *Foxk2*-deficient mice. Preliminary omics analysis suggested FOXK2 directly regulates the expression of mitochondrial function-related genes by modulating chromatin accessibility at its binding sites. Our study identifies *FOXK2* as a novel pathogenic gene for congenital myopathy with ptosis and highlights its essential role in skeletal muscle development and mitochondrial homeostasis, offering insights for potential diagnostics and therapies.

**Keywords** Ptosis; FOXK2; Skeletal Muscle Development; Mitochondrial Homeostasis; Coenzyme Q10
**Subject Categories** Genetics, Gene Therapy & Genetic Disease; Musculoskeletal System

## Introduction

Congenital ptosis [MIM:178300 and 300245] is characterized by the drooping of the upper eyelid, present at birth or within the first year of life (Finsterer, 2003; SooHoo et al, 2014). This condition primarily results from the underdevelopment or dysfunction of the levator palpebrae muscle, a skeletal muscle responsible for lifting the upper eyelid, or from abnormalities in its innervation (SooHoo et al, 2014). Congenital ptosis affects the vertical dimension of the palpebral fissure and can lead to partial or complete occlusion of vision (Edmunds et al, 1998; Finsterer, 2003). As a result, it may cause significant visual impairments, including myopia, amblyopia, and strabismus, as well as associated facial abnormalities. These issues can contribute to psychological disorders and reduce the quality of life for patients and their families. The prevalence of congenital ptosis is approximately 19.9 per 100,000 children worldwide, with a notably higher incidence rate of 1.8 per 1000 individuals in China (Hu, 1987). Despite being relatively uncommon, the impact of the condition on vision and appearance necessitates a deeper understanding of its underlying causes and potential treatments.

Genetic factors play a crucial role in the pathogenesis of congenital ptosis, with an estimated heritability of 0.75 (Vestal et al, 1990). Research has identified chromosomal abnormalities (Engle et al, 1997; McMullan and Tyers, 2001; Nakashima et al, 2008) and specific genetic mutations as significant contributors to the condition (Fazeli et al, 2017). Two primary candidate genes, *ZFHX4* (McMullan et al, 2002; Nakashima et al, 2008) [MIM:606940] and *COL25A1* (Khan and Al-Mesfer, 2015) [MIM:610004], have been implicated in isolated congenital ptosis. These genes encode proteins essential for myogenic differentiation (Gonçalves et al, 2019; Hemmi et al, 2006). Additionally, *ZFHX4* is located in the chromosomal deletion region associated with 8q21.11 deletion syndrome [MIM:614230], which is characterized by congenital ptosis, hypotonia, and short stature (Fontana et al, 2021; Palomares et al, 2011). Mutations in *COL25A1* are also linked to congenital fibrosis of the extraocular muscles [MIM:616219], resulting in clinical manifestations such as congenital ptosis and abnormal development of extraocular muscles, which can impair eye movement

[1]Key Laboratory of Metabolism and Molecular Medicine, Ministry of Education, Department of Biochemistry and Molecular Biology, School of Basic Medical Sciences; ENT institute, Department of Facial Plastic and Reconstructive Surgery, Eye & ENT Hospital; Institute of Medical Genetics & Genomics, Fudan University, Shanghai 200032, China. [2]Dalian Women and Children's Medical Group Neurology Department, Dalian 116012, China. [3]The Central Hospital of Wuhan, Tongji Medical College, Huazhong University of Science and Technology, Wuhan 430030, China. [4]CAS Key Laboratory of Nutrition, Metabolism and Food Safety, Shanghai Institute of Nutrition and Health, University of Chinese Academy of Sciences, Chinese Academy of Sciences, Shanghai 200031, China. [5]Department of Immunology, School of Basic Medical Sciences, Zhengzhou University, Zhengzhou, Henan 450001, China. [6]Henan Eye Institute, Henan Academy of Innovations in Medical Science, Zhengzhou, Henan 450000, China. [7]These authors contributed equally: Peixuan Wu, Nan Song, Yang Xiang. ✉E-mail: duanma@fudan.edu.cn; ty.zhang2006@aliyun.com; ty_zhang@fudan.edu.cn; haobt123@163.com; haobt123@zzu.edu.cn; mj19815208@yeah.net; maj14@fudan.edu.cn

and potentially cause blindness (Munezane et al, 2019). This suggests that mutations in genes responsible for congenital ptosis can lead to more severe disorders related to skeletal muscle development.

Congenital ptosis is classified as a localized myopathy (Brodsky, 2000) and is part of a broader phenotype spectrum of severe congenital myopathies, including centronuclear myopathy (Chen et al, 2015) [MIM:160150] and myotonia congenita (Odrzywolski et al, 2013) [MIM:160800]. Pathogenic mutations in genes such as *MTM1* (Bevilacqua et al, 2009) [MIM:300415] and *DMPK* (Kaliman and Llagostera, 2008) [MIM:605377], associated with these myopathies, lead to dysplasia of skeletal muscles, including the levator palpebrae muscle. Interestingly, ptosis often appears earlier than other symptoms in these conditions (Kornblum et al, 2013), making it an early indicator of more extensive skeletal muscle development disorders.

Despite recognizing the genetic basis of congenital ptosis, research into its genetic underpinnings remains limited. This study aims to deepen our understanding of the genetic factors involved by identifying *FOXK2* [MIM:147685] as a novel pathogenic gene linked to congenital myopathy associated with ptosis. Through whole-exome sequencing (WES) of clinical families and functional studies in animal and cell models, we reveal the role of FOXK2 in regulating skeletal muscle development. FOXK2 maintains mitochondrial function, which is crucial for the development of muscle stem cells (MuSCs). This discovery provides a new perspective on the genetic etiology of congenital ptosis and has broader implications for disorders related to skeletal muscle development. Ultimately, these insights pave the way for improved strategies in the prevention and treatment of these conditions.

# Results

## Identification of *FOXK2* mutations in pedigrees of congenital myopathy associated with ptosis

In clinical practice, we investigated a four-generation pedigree with isolated congenital ptosis, which followed an autosomal dominant inheritance pattern (Fig. 1A). Analysis of eyelid muscle (EM) tissue from the proband revealed disorganized muscle fiber arrangement (Fig. 1B). Additionally, the proband's EM showed reduced MyHC expression compared to the control, indicating impaired muscle development (Fig. 1C). To determine the genetic basis, we performed WES on eight individuals from the pedigree, including five affected and three unaffected individuals. Our comprehensive bioinformatics analysis and literature review identified a mutation (c.643 C > T: p.R215W) in the *FOXK2* gene among the affected individuals, suggesting its potential pathogenic role in this pedigree (Fig. 1A). Furthermore, FOXK2 expression in the EM of the proband is diminished (Fig. 1D).

Since no previous reports linked *FOXK2* with congenital ptosis and recognizing that genes associated with ptosis might also contribute to developmental abnormalities in other skeletal muscle groups, we conducted a detailed examination of *FOXK2* mutations in WES data from congenital myopathy cases with ptosis but no known pathogenic gene. In this investigation, we identified five distinct mutations within *FOXK2* (c.164 C > T:p.T55I, c.1231 C > G:p.P411A, c.1436 C > T:p.A479V, c.1631 G > A:p.R544Q, c.1650 G > C:p.Q550H) across five different pedigrees. We further assessed these mutations through in vitro

western blot experiments. The T55I, R215W, P411A, R544Q, and Q550H mutations resulted in reduced FOXK2 protein expression, whereas the A479V mutation did not, leading to its exclusion from further analysis (Figs. 1E,F and EV1). Sanger sequencing confirmed that all five mutations exhibited phenotypic-genotypic co-segregation within each pedigree (Fig. 1A,E,F). Additional details regarding the five pedigrees are provided in Table 1.

FOXK2 is a highly conserved transcription factor that features a DNA-binding forkhead box domain (FOX), a nuclear localization sequence (NLS), and a protein-protein interaction forkhead-association domain (FHA) (Kang et al, 2022). We have highlighted the positions of amino acid changes caused by these mutations on the protein's structural representation (Fig. 1G). The five mutations in *FOXK2* lead to alterations in evolutionarily conserved amino acids across different species and are predicted to potentially have a pathogenic impact according to multiple prediction tools (Fig. 1H; Table 2). These results strongly suggest that *FOXK2* may be a novel pathogenic gene linked to disorders of skeletal muscle development.

## The knockdown of *foxk2* in zebrafish results in skeletal muscle dysplasia

Zebrafish serve as an effective model for studying congenital myopathies due to their similarity to human skeletal muscle structure and contraction characteristics (Goody et al, 2017; Yong et al, 2024). Their short reproductive cycle and transparent bodies also facilitate observation. To investigate the effects of FOXK2 abnormalities on skeletal muscle development, we used the transgenic zebrafish strain *Tg(-1.9mylpfa:EGFP)*, which has fluorescently labeled skeletal muscle structures (Ma et al, 2019). We downregulated the homologous gene *foxk2* in fertilized eggs by microinjecting morpholinos (MO) ATG-MO and I2E3-MO targeting the *foxk2* gene (Fig. 2A,B). Compared to the control group, larvae in the downregulated groups showed defects or mortality (Fig. 2C). At 72-hours post fertilization (hpf), the survival rates of larvae in the downregulated groups were significantly lower, at 48.33% and 10.4%, respectively (Fig. 2D).

Following *foxk2* downregulation, larvae exhibited shortened body length, a bent trunk (Figs. 2E and EV2A), disorganized skeletal muscle fiber arrangement in the trunk, and loss of skeletal muscles in the craniofacial region (Fig. 2F,G). Zebrafish demonstrate typical motor behaviors, and we assessed the functionality of their skeletal muscles through touch-evoked escape behavior assays at 4–5 days post fertilization (dpf) after the development of the swim bladder (Smith et al, 2013). Downregulation of *foxk2* led to significant reductions in movement distance, speed, maximum acceleration, and overall mobility of larvae compared to the control group (Figs. 2H–K and EV2B).

In summary, FOXK2 deficiency results in abnormal skeletal muscle development and impaired motor ability in zebrafish, underscoring the crucial role of FOXK2 in proper skeletal muscle development of zebrafish.

## Deletion of the *Foxk2* gene in muscle stem cells results in skeletal muscle dysplasia of mice

Myogenesis, the process of muscle formation, involves the differentiation of MuSCs into myoblasts and myocytes, followed by the fusion of myocytes to form multinucleated myofibers. In

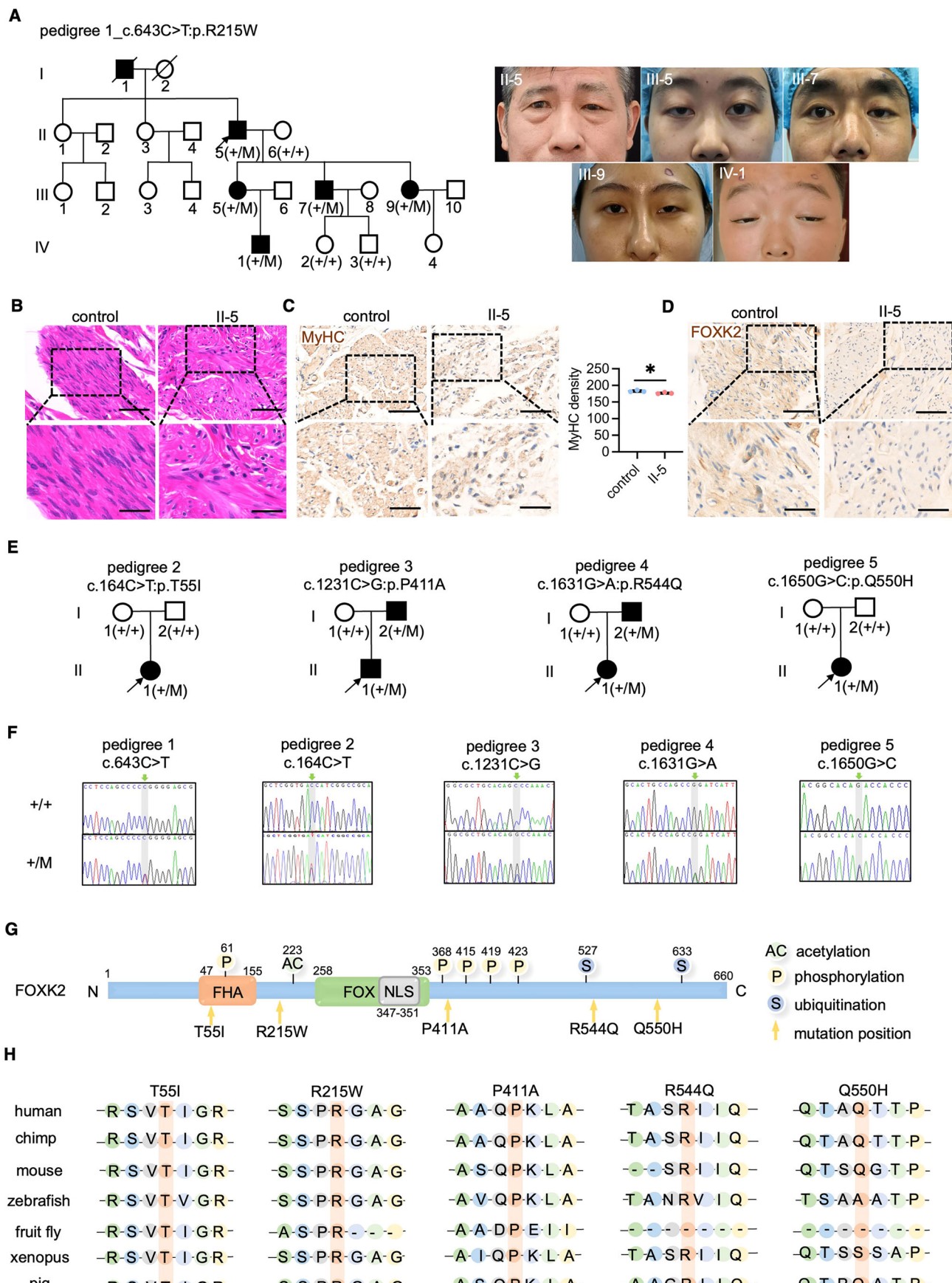

**Figure 1. Identification of *FOXK2* mutations in pedigrees of congenital ptosis and associated myopathy.**

(A) Pedigree illustrating isolated congenital ptosis with a heterozygous *FOXK2* mutation (c.643 C > T:p.R215W). Photographs of affected individuals are included. Black-filled symbols denote affected members. '+' indicates the wild-type *FOXK2* allele, while 'M' represents the mutated allele. The black arrow points to the proband. Genotypes of some individuals remain unspecified due to the unavailability of genomic DNA. (B) H&E staining of eyelid muscle tissues from a control individual and the proband (II-5) in the pedigree described in (A). The zoomed area (scale bars: 50 μm) provides an enlarged view of the boxed region in the left image (scale bars: 100 μm). (C) IHC staining of MyHC in the eyelid muscle samples of a control individual and the proband (II-5) in the pedigree described in (A) ($n = 3$ area per group). The zoomed area (scale bars: 50 μm) offers an enlarged view of the boxed region in the left image (scale bars: 100 μm). Quantitative analysis was performed using ImageJ software, with results expressed by InDen/Area. *P* value: *$P = 0.0448$. (D) IHC staining of FOXK2 in the eyelid muscle samples of a control individual and the proband (II-5) in the pedigree described in (A). The zoomed area (scale bars: 50 μm) offers an enlarged view of the boxed region in the left image (scale bars: 100 μm). (E) Pedigrees depicting congenital myopathy associated with ptosis, carrying various heterozygous *FOXK2* mutations (c.164 C > T:p.T55I; c.1231 C > G:p.P411A; c.1631 G > A:p.R544Q; c.1650 G > C:p.Q550H). '+' denotes the wild-type *FOXK2* allele, and 'M' represents the mutated allele. The black arrow identifies the proband. (F) Sanger sequencing diagrams of the *FOXK2* mutation sites in healthy individuals and patients across five pedigrees. Green arrows in the upper panels indicate the mutation sites, highlighted in gray. (G) Diagram of the FOXK2 protein showing the locations of the mutations. (H) Species conservation analysis of FOXK2 amino acids p.T55, p.R215, p.P411, p.R544, and p.Q550. Data were analyzed by Student's t test. All error bars represent mean ± standard deviation. H&E hematoxylin-eosin staining, FHA forkhead-associated domain, FOX forkhead DNA-binding domain, NLS nuclear localization sequence, IHC immunohistochemistry. Source data are available online for this figure.

mice, skeletal muscle development occurs in three key stages: the formation of primary myofibers (embryonic day (E) 10.5-E12.5), the formation of secondary myofibers (E14.5-E17.5) (Chal and Pourquié, 2017; Dos Santos et al, 2023), and the extensive proliferation and fusion of MuSCs that contribute myonuclei to growing myofibers postnatally, continuing until muscle maturation at 8 weeks of age (Bachman et al, 2018; Dos Santos et al, 2023). We examined the skeletal muscles of mouse hindlimbs at various developmental stages and noted increased expression of *Foxk2* during these critical periods (Appendix Fig. S1A). This observation suggests the association of FOXK2 with mouse skeletal muscle development.

Then, to conform this role of FOXK2, we generated MuSCs-specific *Foxk2* conditional knockout mice by crossing *Foxk2*^fl/fl^ with *Myod1*-Cre (Appendix Fig. S1B,C) (White et al, 2010), prompted by the decrease of FOXK2 expression in the proband's EM from pedigree 1. The *Foxk2*^fl/fl^; *Myod1*-Cre mice were viable, and all offspring were born at expected Mendelian ratio, irrespective of gender (Appendix Fig. S1D). We verified the effectiveness of *Foxk2* deletion in various skeletal muscles, including gastrocnemius (Gas), tibialis anterior (TA), diaphragm (Dp), intercostal muscle (IM), tongue muscle (TM), and EM, through immunohistochemistry (IHC) staining (Appendix Fig. S1E). Heterozygous mice (*Foxk2*^fl/+^; *Myod1*-Cre) did not show any noticeable abnormal phenotypes. In contrast, all homozygous mice (*Foxk2*^fl/fl^; *Myod1*-Cre) exhibited postnatal short stature, significantly lower body weight, and reduced muscle sizes in the TA and Gas. Additionally, the ratio of Gas weight to body weight was notably decreased compared to their *Foxk2*^fl/fl^ littermates at 8 weeks of age (Fig. 3A–C). The abnormal skeletal muscle development also affected muscle endurance, as evidenced by a significantly reduced hanging endurance time in the homozygous mice (Fig. 3D).

During skeletal muscle development, myofiber fusion is associated with the movement of myonuclei from the center to the periphery (Cadot et al, 2015). An increased number of myofibers with central nuclei is a marker of skeletal muscle dysplasia (Haahr et al, 2022; Mazzotti and Coletti, 2016). To evaluate this, we examined the presence of central nuclei myofibers in typical skeletal muscles of mice at various developmental stages: postnatal day 1, 4 weeks, and 8 weeks. In control mice, we observed central nuclei myofibers at day 1 due to the immaturity of skeletal muscle. In contrast, *Foxk2*^fl/fl^; *Myod1*-Cre mice displayed a significantly higher proportion of central nuclei myofibers in the

Gas, TA, and IM (Figs. 3E and EV3A). This increase was more pronounced and widespread across all skeletal muscles at 4 and 8 weeks of age, when skeletal muscle maturation was expected (Figs. 3E and EV3A).

Additionally, compared to *Foxk2*^fl/fl^ mice, *Foxk2*^fl/fl^; *Myod1*-Cre mice showed reduced expression of the muscle maturation protein MyHC in their skeletal muscles (Figs. 3F and EV3B). Analysis of TA sections, with myofiber boundaries marked by an antibody against Laminin, revealed a higher proportion of small myofibers and a lower proportion of large myofibers in *Foxk2*^fl/fl^; *Myod1*-Cre mice (Fig. 3G). This finding indicates a significantly smaller average myofiber size. We also generated germline *Foxk2* knockout mice, which followed the expected Mendelian birth ratio (Appendix Fig. S2). *Foxk2*^−/−^ mice exhibited skeletal muscle developmental abnormalities similar to those observed in *Foxk2*^fl/fl^; *Myod1*-Cre mice when compared to their *Foxk2*^+/+^ littermates (Appendix Fig. S3). These results suggest that *Foxk2* knockout leads to skeletal muscle dysplasia in mice, highlighting FOXK2 as a crucial regulator of skeletal muscle development.

## *Foxk2* knockout disrupts mitochondrial homeostasis in mouse muscle stem cells

The process of MuSCs relies heavily on substantial mitochondrial energy production, highlighting the critical role of mitochondrial homeostasis in skeletal muscle development (Remels et al, 2010). Maintaining mitochondrial homeostasis depends on the structural integrity of mitochondria and the balance of mitochondrial dynamics, including fusion and fission (Romanello and Sandri, 2021). Disruptions in mitochondrial homeostasis can lead to skeletal muscle disorders, characterized by abnormalities such as altered mitochondrial membrane potential, reduced ATP production, and increased levels of reactive oxygen species (ROS) (Chen et al, 2023).

FOXK2 has been shown to regulate mitochondrial metabolism in adipocytes (Sakaguchi et al, 2019; Sukonina et al, 2019), but its impact on mitochondrial homeostasis during skeletal muscle development has not been thoroughly explored. To investigate this, we examined mitochondrial morphology and function in the TA tissue of 8-week-old mice. Transmission electron microscope (TEM) analysis revealed that mitochondria in the TA of *Foxk2*^fl/fl^ mice were intact. In contrast, *Foxk2*^fl/fl^; *Myod1*-Cre mice exhibited damaged mitochondria with indistinct cristae (Fig. 4A), similar to

**Table 1. The detailed information of FOXK2 mutation and proband information.**

| Pedigree | Chromosomal position | cDNA alteration | Amino acid alteration | Mutation zygosity | Age of proband | Proband phenotype |
|---|---|---|---|---|---|---|
| 1 | chr17:80525958 | c.643 C > T | p.Arg215Trp | Het | 62 years | Isolated congenital ptosis |
| 2 | chr17:80477928 | c.164 C > T | p.Thr55Ile | Het | 8 years and 11 months | Congenital ptosis, skeletal muscle atrophy, growth delay, motor delay, global developmental delay |
| 3 | chr17:80542016 | c.1231 C > G | p.Pro411Ala | Het | 2 years and 10 months | Congenital ptosis, short stature, small for gestational age, skeletal muscle atrophy |
| 4 | chr17:80544993 | c.1631 G > A | p.Arg544Gln | Het | 5 years and 5 months | Congenital ptosis, skeletal muscle atrophy, global developmental delay, growth abnormality, abnormality of limbs |
| 5 | chr17:80545012 | c.1650 G > C | p.Gln550His | Het | 2 years | Congenital ptosis, hypertonia, motor delay |

Information of FOXK2 mutations in five pedigrees (one isolated congenital ptosis and four congenital myopathy associated with ptosis), as well as the age and phenotype information of proband. The reference of FOXK2 transcript is GenBank: NM_004514.4. Het heterozygous.

the mitochondrial morphology observed in the EM of individuals with congenital ptosis (Surve et al, 2019). Additionally, TA sections from Foxk2$^{fl/fl}$; Myod1-Cre mice showed elevated ROS levels and decreased ATPase activity, indicating a reduction in myofiber contraction strength (Schiaffino and Reggiani, 2011) (Fig. 4B,C).

We further isolated primary MuSCs from the hindlimb skeletal muscles of mice. Cells from Foxk2$^{fl/fl}$; Myod1-Cre mice showed increased expression of the mitochondrial fission marker DRP1 and decreased expression of the mitochondrial fusion marker OPA1 compared to cells from Foxk2$^{fl/fl}$ mice. This suggests an imbalance in mitochondrial dynamics (Fig. 4D). Foxk2 deficiency also resulted in significant mitochondrial dysfunction in mouse MuSCs, including decreased mitochondrial membrane potential, elevated ROS levels, and reduced ATP production (Fig. 4E–G).

However, the assessment of mitochondrial mass, as indicated by mitochondrial DNA (mtDNA) copy number quantification (Popov, 2020), revealed no significant difference between MuSCs from Foxk2$^{fl/fl}$ and Foxk2$^{fl/fl}$; Myod1-Cre mice. This suggests that Foxk2 knockout does not markedly affect mitochondrial mass (Fig. 4H). Additionally, the reduced mRNA levels of several mitochondria-related genes (Tfb2m, Cycs, and Cox4l) in MuSCs from Foxk2$^{fl/fl}$; Myod1-Cre mice compared to Foxk2$^{fl/fl}$ mice indicate that Foxk2 deficiency partially impairs mitochondrial biogenesis (Fig. 4I). These results underscore the essential role of FOXK2 in maintaining mitochondrial homeostasis during skeletal muscle development.

## FOXK2 deficiency impairs myogenic differentiation and mitochondrial homeostasis in C2C12 cells

The C2C12 cell line, derived from mouse myogenic progenitor cells, is extensively used to study skeletal muscle development (Hiraumi et al, 2016). Upon induction with differentiation medium containing 2% horse serum for 5–8 days, C2C12 cells show decreased expression of MuSCs markers (PAX7, MYOD, MYF5) and increased expression of terminal differentiation markers such as MyHC (encoded by the Myh4 gene) and MYOG (Jiang et al, 2023; Luo et al, 2019). In our study, we induced myogenic differentiation of C2C12 cells in vitro and performed western blot and quantitative real-time polymerase chain reaction (qPCR) analysis. Following differentiation, we observed a significant decline in FOXK2 expression, mirroring the pattern of MuSCs markers (Fig. 5A,B). This finding indicates the importance of FOXK2 during the initial stage of myogenesis. Then, we used the CRISPR/ Cas9 system to delete the Foxk2 gene (FOXK2-KO) in C2C12 cells, employing a single guide RNA (sgRNA) targeting exon 1 of the Foxk2 gene. Additionally, we conducted rescue experiments to assess the functional impact of the FOXK2 mutation (R215W) identified in the pedigree with isolated congenital ptosis (Fig. 1A). For these experiments, we introduced plasmids expressing either wild-type human FOXK2 or the R215W mutation into FOXK2-KO cells (Fig. 5C).

After six days of differentiation, FOXK2-KO cells showed decreased expression levels of Myh4 and Myog compared to control cells. Remarkably, the introduction of wild-type FOXK2 protein into FOXK2-KO cells effectively restored the expression of muscle differentiation marker genes Myh4 and Myog. In contrast, the mutated FOXK2 did not achieve the same rescue effect (Fig. 5D). During terminal differentiation, C2C12 cells fuse to form

**Table 2. Allele frequency and pathogenicity prediction of *FOXK2* mutations.**

| Pedigree | Mutation | Allele frequency | | | | In silico bioinformatics prediction | | |
|---|---|---|---|---|---|---|---|---|
| | | ExAC | ExAC_EAS | gnomAD | gnomAD_EAS | Mutationtaster | PolyPhen2_HDIV | CADD |
| 1 | c.643 C > T p.Arg215Trp | NA | NA | 8.24E-06 | 0 | D (1.0) | B (0.146) | 25.8 |
| 2 | c.164 C > T p.Thr55Ile | NA | NA | NA | NA | D (1.0) | PD (0.885) | 20.7 |
| 3 | c.1231 C > G p.Pro411Ala | 8.51E-05 | 0.001 | 6.52E-05 | 0.0008 | D (1.0) | B (0.112) | 22.7 |
| 4 | c.1631 G > A p.Arg544Gln | 0.0023 | 0.0001 | 0.0021 | 0.0002 | D (1.0) | PD (0.999) | 27.9 |
| 5 | c.1650 G > C p.Gln550His | NA | NA | 3.99E-06 | 0 | P (0.91) | PD (0.966) | 22.8 |

The allele frequency and in silico bioinformatics prediction of *FOXK2* mutations in five pedigrees (one isolated congenital ptosis and four congenital myopathy associated with ptosis).
*ExAC* exome aggregation consortium, *EAS* East Asian, *gnomAD* Genome Aggregation Database, *Polyphen2* polymorphism phenotyping v2, *HDIV* HumDiv model, *CADD* Combined Annotation Dependent Depletion, *NA* not available, *D* damaging, *P* polymorphism, *B* benign, *PD* possible damaging.

multinucleated myotubes, which can be detected by MyHC immunostaining (Braun and Gautel, 2011; Guo et al, 2022). In FOXK2-KO cells, the typical long and well-organized multi-nucleated myotubes observed in control cells were replaced by fewer, shorter, thinner and disorganized myotubes. Importantly, introducing wild-type FOXK2 protein restored normal myotube morphology, whereas the mutated FOXK2 protein did not (Fig. 5E).

Based on our observation that *Foxk2* deletion disrupts mitochondrial homeostasis in mice (Fig. 4), we further investigated mitochondrial phenotypes in C2C12 cells. Deletion of FOXK2 resulted in impaired mitochondrial microstructure, characterized by incomplete mitochondrial membranes and indistinct cristae (Fig. 6A). Increased expression of DRP1 and decreased expression of OPA1 in FOXK2-KO cells indicated excessive mitochondrial fission, leading to highly fragmented mitochondria and impaired function (Figs. 6B,C and EV4A,B). Consistent with the abnormal mitochondrial morphology, FOXK2-KO cells exhibited significant mitochondrial dysfunction, including decreased mitochondrial membrane potential, reduced ATP production, elevated ROS levels, decreased mtDNA copy number, and reduced expression of mitochondria-related genes (Figs. 6D–I and EV4C). Although the wild-type FOXK2 protein effectively restored mitochondrial function in FOXK2-KO cells, the R215W mutant was less effective, only partially restoring mitochondrial function (Fig. 6A–C,F,G).

In summary, FOXK2 deficiency impairs myogenic differentiation and disrupts mitochondrial homeostasis in C2C12 cells. The R215W mutation acts as a loss-of-function mutation, leading to abnormal myogenic differentiation and mitochondrial dysfunction.

## Coenzyme Q10 alleviates skeletal muscle disorders of mice caused by *Foxk2* deficiency by improving mitochondrial function

Coenzyme Q10 (CoQ10) is a critical component of the mitochondrial respiratory chain, playing essential roles in electron transfer, reducing oxidative stress, enhancing mitochondrial function, and promoting energy metabolism(Jing et al, 2015; Perez-Sanchez et al, 2012; Xie et al, 2020). Given the observed mitochondrial dysfunction in *Foxk2*<sup>fl/fl</sup>; *Myod1*-Cre mice (Fig. 4), we administered CoQ10 *via* oral gavage to improve mitochondrial function in skeletal muscle and assess its effects on skeletal muscle development in mice (Fig. 7A). We detected increased CoQ10 concentration in the TA of both *Foxk2*<sup>fl/fl</sup> and *Foxk2*<sup>fl/fl</sup>; *Myod1*-Cre mice

(Fig. 7B). CoQ10 treatment restored ATP levels in the skeletal muscle of *Foxk2*<sup>fl/fl</sup>; *Myod1*-Cre mice to those of the *Foxk2*<sup>fl/fl</sup> group (Fig. 7C). Histological analysis revealed that CoQ10 treatment improved mitochondrial morphology, resulting in a greater number of mitochondria with well-defined membrane structures and intact mitochondrial cristae (Fig. 7D), and reduced ROS levels in the *Foxk2*<sup>fl/fl</sup>; *Myod1*-Cre mice group compared with the vehicle group (Fig. 7E). These findings confirm that CoQ10 significantly improves mitochondrial function in skeletal muscles of *Foxk2*<sup>fl/fl</sup>; *Myod1*-Cre mice. Additionally, CoQ10 treatment decreased the proportion of central nuclei myofibers in the skeletal muscles (Gas, IM, TM, Dp and EM) of *Foxk2*<sup>fl/fl</sup>; *Myod1*-Cre mice compared to the vehicle group (Fig. 7F). Notably, the CoQ10 treatment group exhibited a higher proportion of large myofibers and a lower proportion of small myofibers (Fig. 7G), indicating improved skeletal muscle maturation of *Foxk2*<sup>fl/fl</sup>; *Myod1*-Cre mice.

Overall, all findings suggest that CoQ10 treatment promotes skeletal muscle development in mice with *Foxk2* deficiency by improving mitochondrial functions in skeletal muscle.

## FOXK2 regulates mitochondrial function by directly transcriptional regulating *Pdk4* and *Cry1*

As a transcription factor, FOXK2 regulates gene expression through genome-wide DNA binding (Sukonina et al, 2019). To investigate its molecular mechanism, we performed chromatin immunoprecipitation sequencing (ChIP-seq) to identify the genome-wide binding sites of FOXK2 in C2C12 cells. We identified 853 peaks, the majority (64.24%) of which were located in promoter regions (Fig. 8A; Dataset EV1). These peaks contained a putative FOXK2 binding motif (Fig. 8B), consistent with previous reports (Chen et al, 2016; Ji et al, 2012). Genes with FOXK2-bound promoters were considered direct target genes. Gene Ontology (GO) analysis of these targets revealed significant enrichment in categories related to mitochondrial function, such as mitochondrial membrane organization (crucial for ATP production) (Klecker and Westermann, 2021) and cytochrome c release from mitochondria (associated with ROS levels) (Petrosillo et al, 2003) (Figs. 8C and EV5A).

Transcription factors can also influence gene expression by modulating chromatin accessibility (60). Therefore, we performed assay of transposase accessible chromatin sequencing (ATAC-seq) to assess the change of chromatin accessibility in FOXK2-KO C2C12 cells. We identified 3134 regions with altered chromatin accessibility 3109 regions showing increased accessibility, while 25

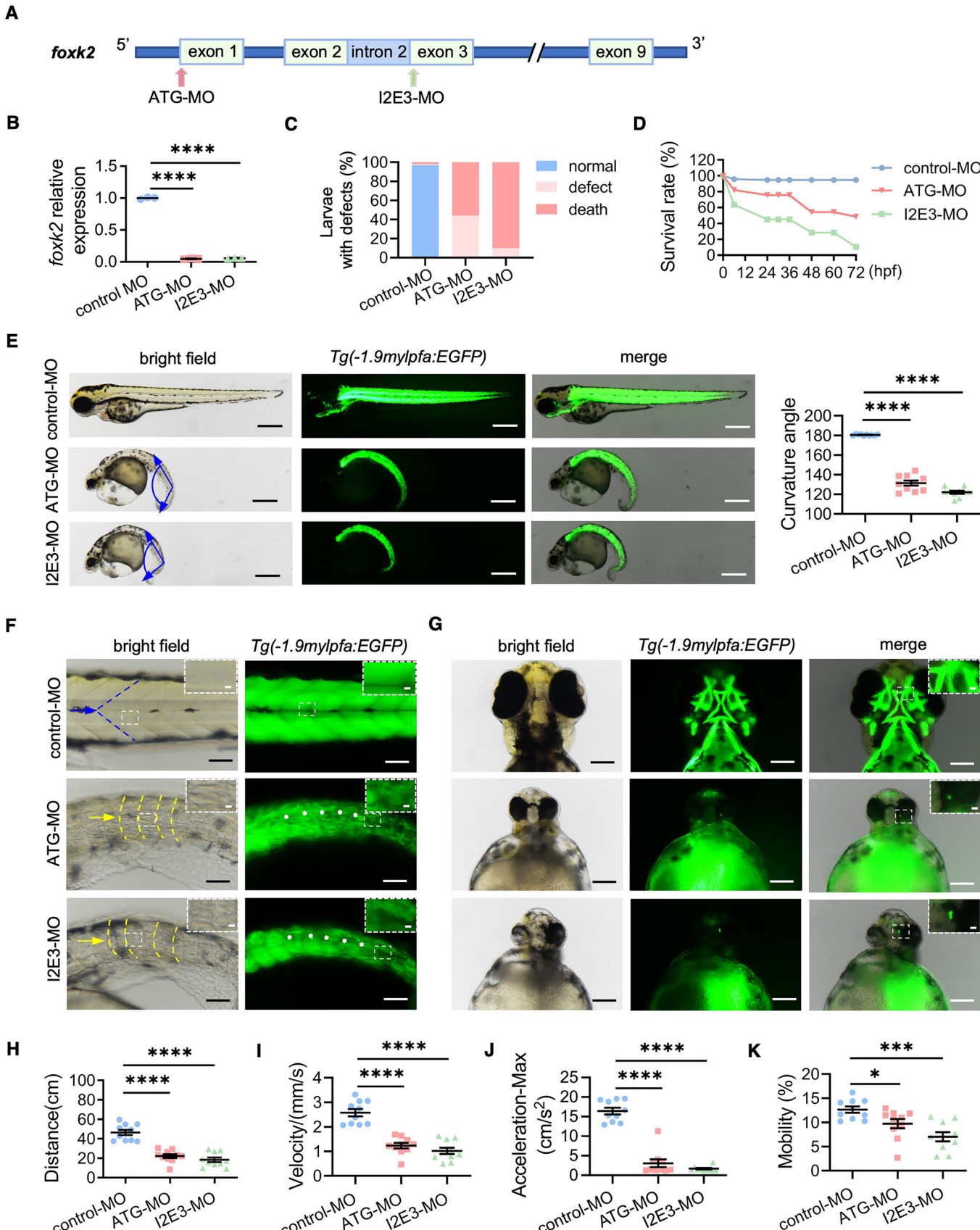

◄ **Figure 2.** *foxk2* knockdown impaired skeletal muscle development of zebrafish.

(**A**) Structure of the zebrafish *foxk2* gene. The zebrafish *foxk2* gene was targeted using specific MOs to inhibit transcription at the start codon (ATG-MO) and proper splicing of exon 3 (I2E3-MO). (**B**) Quantitative of *foxk2* expression level by qPCR. Larvae were collected at 72 hpf ($n = 3$ repeats). $P$ value: ****$P < 0.0001$. (**C**) Percentage of larvae exhibiting defect or mortality. The control-MO group included 379 larvae, the ATG-MO group included 296 larvae, and the I2E3-MO group included 375 larvae. (**D**) Survival rate of larvae over a 72-hpf time course in control-MO, ATG-MO, and I2E3-MO groups. (**E**) Gross morphology of *Tg(-1.9mylpfa:EGFP)* larvae at 72-hpf, along with the quantification of curvature angle of larvae ($n = 10$ larvae for each group). The angle formed between the blue arrows indicates the degree of body curvature in the larvae. Scale bars: 500 μm. $P$ value: ****$P < 0.0001$. (**F**) Body muscle morphology of *Tg(-1.9mylpfa:EGFP)* larvae at 72-hpf. Blue and yellow arrows denote normal and defective somites, respectively. The white asterisk indicates an abnormal trunk orientation. The inset in the upper right corner shows a magnified view of the boxed area (scale bars: 10 μm for the inset, 50 μm for the full image). (**G**) Cranial facial muscles of *Tg(-1.9mylpfa:EGFP)* larvae at 72-hpf. The inset in the upper right corner shows a magnified view of the boxed area (scale bars: 40 μm for the inset, 200 μm for the full image). (**H–K**) Statistical analysis of (**H**) movement distance ($P$ value: ****$P < 0.0001$), (**I**) velocity (****$P < 0.0001$), (**J**) maximum acceleration (****$P < 0.0001$) and (**K**) mobility (percentage of actual moving distance to total distance) (***$P = 0.0001$; *$P = 0.0231$) in larvae at 5-dpf ($n = 10$ larvae for each froup). Data were analyzed by Student's t test. All error bars indicate the mean ± standard deviation. MO morpholino, hpf hours post fertilization, qPCR quantitative real-time polymerase chain reaction, dpf days post fertilization. Source data are available online for this figure.

regions exhibiting decreased accessibility (Fig. 8D; Dataset EV2). A heatmap of these regions confirmed increased chromatin accessibility at FOXK2 binding sites following its knockout (Fig. 8E), underscoring FOXK2's role in chromatin remodeling. The gained peaks were associated with processes such as mitochondrial transport and regulation of mitochondrial organization (Fig. 8F).

An integrated analysis of the promoter-associated genes from ChIP-seq, the upregulated peaks from ATAC-seq, and mitochondrial function related gene set from the MitoCarta3.0 dataset (https://www.broadinstitute.org/mitocarta/mitocarta30-inventory-mammalian-mitochondrial-proteins-and-pathways) revealed 19 mitochondrial function related genes (Fig. 8G; Dataset EV3). To assess FOXK2's regulatory role on these 19 genes, we performed RNA sequencing (RNA-seq) on FOXK2-KO C2C12 cells, identifying 910 downregulated and 1,107 upregulated differentially expressed genes (DEG) (Figs. 8H and EV5B; Dataset EV4). Among the 19 mitochondrial function related genes, pyruvate dehydrogenase kinase 4 (*Pdk4*) and cryptochrome circadian regulator 1 (*Cry1*) were differentially expressed in FOXK2-KO cells (Table EV1). *Pdk4* codes a mitochondrial protein involved in regulating mitochondrial dynamics (Thoudam et al, 2022), while *Cry1* is a key circadian clock regulator that modulates ROS levels (Ma et al, 2024). We observed that FOXK2 occupies the promoter regions of *Pdk4* and *Cry1*, with *Pdk4* being upregulated and *Cry1* downregulated upon FOXK2 knockout (Fig. 8I). We also detected upregulation of *PDK4* and downregulation of *CRY1* in the EM tissue of the proband from pedigree 1 (Fig. EV5C). These findings were further validated in FOXK2-KO C2C12 cells, as well as in TA tissues and MuSCs isolated from *Foxk2*^fl/fl; *Myod1*-Cre mice (Figs. 8J and EV5D–F). Knockdown of PDK4 and overexpression of CRY1 in FOXK2-KO cells restored ATP production, further confirming the correlation between *PDK4* / *CRY1*, genes directly regulated by FOXK2, and the maintenance of mitochondrial function (Fig. EV5G,H).

Collectively, these findings suggested that FOXK2 may regulate mitochondrial homeostasis in skeletal muscle by affecting the expression of mitochondrial function related genes, *Pdk4* and *Cry1*, through direct binding to their promoter regions and influencing chromatin accessibility.

## Discussion

Here, we identified *FOXK2* as a novel pathogenic gene linked to congenital myopathies associated with ptosis. Our results showed that deleting *Foxk2* in MuSCs of mice led to abnormal mitochondrial function of MuSCs and severe impairments in skeletal muscle development, highlighting its essential role in this process. Notably, CoQ10 treatment was found to enhance mitochondrial performance and ameliorate skeletal muscle development disorders in these mice. Additionally, FOXK2 was shown to directly regulate mitochondrial function related gene *Pdk4* and *Cry1* and affect chromatin accessibility at its binding sites. These findings enhance our understanding of the genetic factors involved in skeletal muscle formation and provide a new conceptual framework for genetic diagnostics and prenatal strategies aimed at preventing congenital myopathies with ptosis. Furthermore, CoQ10 treatment offers a promising avenue for the treatment of these conditions associated with *FOXK2* mutation.

FOXK2 is a member of the FOX transcription factor family, known for its role in regulating various biological processes, particularly skeletal muscle development. Within this family, other members, such as the FOXO proteins, are crucial for overseeing skeletal muscle development by controlling the differentiation, growth, and homeostasis of skeletal muscle cells (Braun and Gautel, 2011). *FOXL2* [MIM:605597] has been identified as the causative gene for blepharophimosis, ptosis, and epicanthus inversus syndrome [MIM:110100] (Heude et al, 2015). Mutations in *FOXL2* lead to impaired eyelid elevation and developmental anomalies in skeletal muscle (Tucker, 2022; Zhang et al, 2011). FOXK2, part of the evolutionarily conserved FOX class K family, is recognized as a key transcriptional regulator of cell homeostasis (van der Heide et al, 2015). It plays a role in priming lineage-specific genes in human embryonic stem cells, aiding their activation during differentiation (Ji et al, 2021). Although the exact connection between FOXK2 and skeletal muscle development is not fully understood, indirect evidence suggests a possible link. For instance, FOXK1, a homolog of FOXK2, regulates the myogenic progenitor cell population essential for skeletal muscle regeneration (Meeson et al, 2007). FOXK1 plays a critical role in the proliferation and differentiation of MuSCs following muscle injury and serves as a molecular marker for the quiescent myogenic progenitor cell population (Garry et al, 2000; Hawke et al, 2003). Additionally, a documented case describes a de novo 0.5 Mb triplication (partial tetrasomy) of chromosome 17q25.3 in a patient with generalized muscular hypotonia and short stature. Genetic analysis indicates that the chromosomal breakpoint occurs within the *FOXK2* gene (Hackmann et al, 2013), which could affect its functionality.

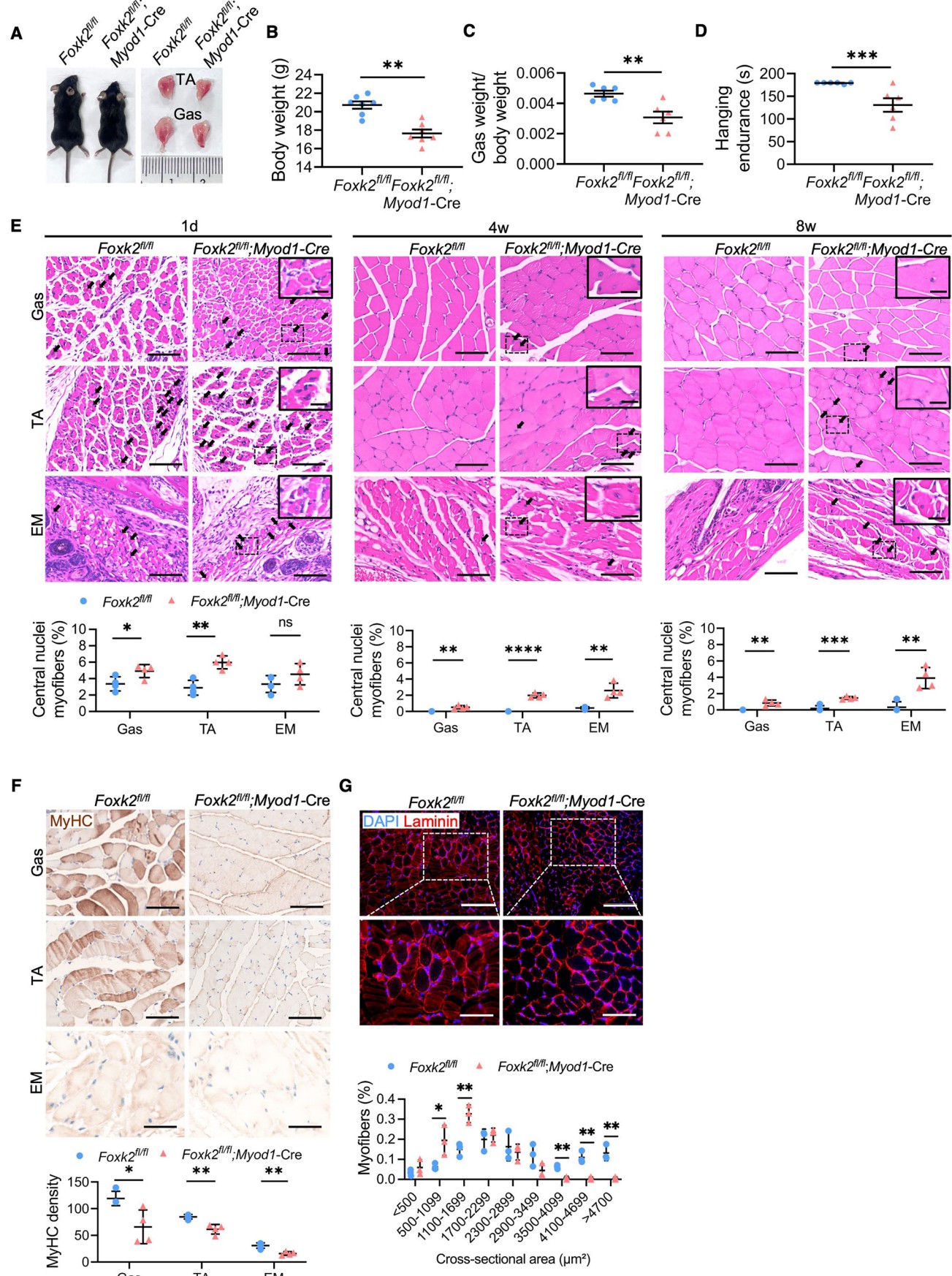

**Figure 3.** *Foxk2* deficiency in muscle stem cells results in skeletal muscle dysplasia in mice.

(A) Representative images of *Foxk2*^fl/fl^ and *Foxk2*^fl/fl^; *Myod1*-Cre littermates at 8 weeks, including their TA and Gas muscle. (B) Body weight of *Foxk2*^fl/fl^ and *Foxk2*^fl/fl^; *Myod1*-Cre mice at 8 weeks (n = 7 mice for each group). P value: **P = 0.0020. (C) Ratio of Gas muscle weight to body weight in *Foxk2*^fl/fl^ and *Foxk2*^fl/fl^; *Myod1*-Cre mice at 8 weeks (n = 6 mice for each group). P value: **P = 0.0047. (D) Hanging endurance time measured by the hanging endurance test in *Foxk2*^fl/fl^ and *Foxk2*^fl/fl^; *Myod1*-Cre mice at 8 weeks (n = 6 mice for each group). P value: ***P = 0.0003. (E) H&E staining illustrating histological aspects and the percentage of central nuclei myofibers in skeletal muscles (Gas, TA, and EM) of *Foxk2*^fl/fl^ and *Foxk2*^fl/fl^; *Myod1*-Cre littermates at postnatal 1 day, 4 weeks, and 8 weeks (n = 4 areas). Black arrows indicate central nuclei. The inset in the upper right corner (scale bars: 25 μm) is an enlarged view of the boxed area in the main image (scale bars: 100 μm), highlighting central nuclei myofibers. P value: 1 d: Gas, *P = 0.0389; TA, **P = 0.0019; EM, ns-P = 0.1708. 4w: Gas, **P = 0.0017; TA, ****P < 0.0001; EM, **P = 0.0031. 8w: Gas, **P = 0.0030; TA, ***P = 0.0006; EM, **P = 0.0026. (F) IHC staining of MyHC in the skeletal muscles (Gas, TA, and EM) of *Foxk2*^fl/fl^ and *Foxk2*^fl/fl^; *Myod1*-Cre littermates at 8 weeks (n = 4 areas). MyHC was quantified by InDen/Area using ImageJ software. Scale bars: 100 μm for Gas and TA; 50 μm for EM. P value: Gas, *P = 0.0212; TA, **P = 0.0033; EM, **P = 0.0029. (G) IF staining of Laminin was the TA of *Foxk2*^fl/fl^ and *Foxk2*^fl/fl^; *Myod1*-Cre littermates at 8 weeks. The distribution of cross-sectional area of myofibers was calculated (n = 3 areas). The zoomed area (scale bars: 100 μm) is an enlarged view of the boxed area in the main image (scale bars: 200 μm). P value: 500-1099, *P = 0.0401; 1100-1699, **P = 0.0049; 3500-4099, **P = 0.0014; 4100-4699, **P = 0.0041; >4700, **P = 0.0047. Data were analyzed by Student's t test. All error bars indicate mean ± standard deviation. TA tibialis anterior, Gas gastrocnemius, d day, w week, EM eyelid muscle, H&E hematoxylin-eosin staining, IHC immunohistochemistry, IF immunofluorescence. Source data are available online for this figure.

Our research confirms the significant role of FOXK2 in skeletal muscle development and highlights its importance within the broader context of FOX family transcription factors in skeletal muscle biology. In our clinical study, we observed five pedigrees with distinct *FOXK2* mutations, each displaying a wide range of clinical phenotypes. Some individuals show isolated ptosis, while others present with additional skeletal muscle developmental disorders of varying severity. The effects of mutations at different positions within *FOXK2* can vary, affecting protein stability and functionality to different extents. For instance, the T55I mutation in the FHA domain may disrupt FOXK2's interactions with other proteins. However, the functional consequences of other mutations on *FOXK2* remain unclear due to the unknown roles of the domains where these mutations occur. Moreover, certain mutations might initiate a compensatory cellular response, upregulating associated genes to counterbalance the loss of function (Rossi et al, 2015). Additionally, gene products with dominant negative variants could potentially interfere with the normal function of wild-type alleles (Yamashita et al, 2003). These factors may contribute to the clinical heterogeneity observed among different *FOXK2* mutations. Further investigation is crucial to clarify the specific impacts of these *FOXK2* mutations through functional experiments.

FOXK1 is a conserved homolog of FOXK2, and both proteins play dual roles of transcriptional activation and repression, sharing overlapping functions in regulating a common set of genes (Katoh and Katoh, 2004; Sukonina et al, 2019). However, FOXK2 and FOXK1 do not have entirely redundant functions (Yu et al, 2022). Their expression patterns during mouse skeletal muscle development and C2C12 cell myogenic differentiation differ. FOXK2 is expressed at an earlier stage than FOXK1 (Appendix Fig. S4), which suggests that FOXK2 may function earlier in skeletal muscle development. To date, no clinical cases have been documented where mutations in the *FOXK1* [MIM:616302] gene cause skeletal muscle developmental disorders. Further research is needed to clarify the complex relationship between FOXK2 and FOXK1 in skeletal muscle development.

Regarding the phenotype of the mice, it is important to note that while *Foxk2* homozygous knockout mice showed developmental abnormalities in skeletal muscles throughout the body, including the eyelids, they did not exhibit a ptosis phenotype. This lack of ptosis may result from the infrequent use of eyelids in mice (Doughty, 2001; Kaminer et al, 2011), which could minimize the pathological impact on their movement.

Mutations in mitochondrial genes, such as *MT-ATP6* (Burrage et al, 2014), and nuclear genes, such as *MGME1* (Kornblum et al,

2013) [MIM:615076] and *RRM2B* (Keshavan et al, 2020) [MIM:604712], which are essential for mitochondrial function, can cause mitochondrial myopathy [MIM:500009], sometimes accompanied by ptosis. FOXK2 plays a crucial role in regulating aerobic glycolysis in adipocytes (Sukonina et al, 2019), mitochondrial metabolism in hepatocytes (Sakaguchi et al, 2019), and mitochondrial respiration in lung epithelial cells (El-Mergawy et al, 2024). In our study, we observed skeletal muscle dysplasia in *Foxk2* homozygous knockout mice, attributed to mitochondrial dysfunction. This observation aligns with the characteristics of congenital mitochondrial myopathy (Garone et al, 2013; Olimpio et al, 2021). Our findings establish a robust model for studying mitochondrial myopathy and suggest a potential role for FOXK2 in its development.

Promising treatments targeting mitochondrial function for mitochondrial myopathy include riboflavin (Carrozzo et al, 2014; Chen et al, 2019), deoxynucleoside (Domínguez-González et al, 2019), and CoQ10 (Gempel et al, 2007). CoQ10 serve as a determinant of muscular strength, with low levels potentially indicating an increased risk of sarcopenia in humans (Fischer et al, 2016). Additionally, CoQ10 acts as an antioxidant without exhibiting cytotoxic effect on muscle cells in mice (Mizobuti et al, 2019). Our observation that CoQ10 treatment improves the phenotype of muscle development disorders in mice with *Foxk2* deficiency provides new insights into its potential treatment prospects in for treating congenital myopathy with ptosis caused by *FOXK2* mutations, highlighting the need for further clinical investigation.

In mouse myogenic progenitor cells, we observed that FOXK2 regulates mitochondrial function related gene expression as a transcription factor. Specifically, FOXK2 transcriptionally activates the expression of *Cry1* and represses the expression of *Pdk4*. FOXK2 has been reported to bind the promoter region of *Pdk4* in adipocytes (Sukonina et al, 2019). Notably, *PDK4* is upregulated in metabolic diseases related to mitochondrial dysfunction, particularly in pathologic muscle conditions associated with defective myogenesis (Kim et al, 2023b). Similarly, *Cry1* is downregulated in muscle tissue of a muscular dystrophy mouse model (Hardee et al, 2021). FOXK2 is known to play dual roles in transcriptional regulation: it acts as a transcriptional activator (Sukonina et al, 2019) and a transcriptional repressors (Bowman et al, 2014), which aligns with our observations. Additionally, we found that FOXK2 knockout results in increased chromatin accessibility, which can enhance binding opportunities for both transcriptional activators and

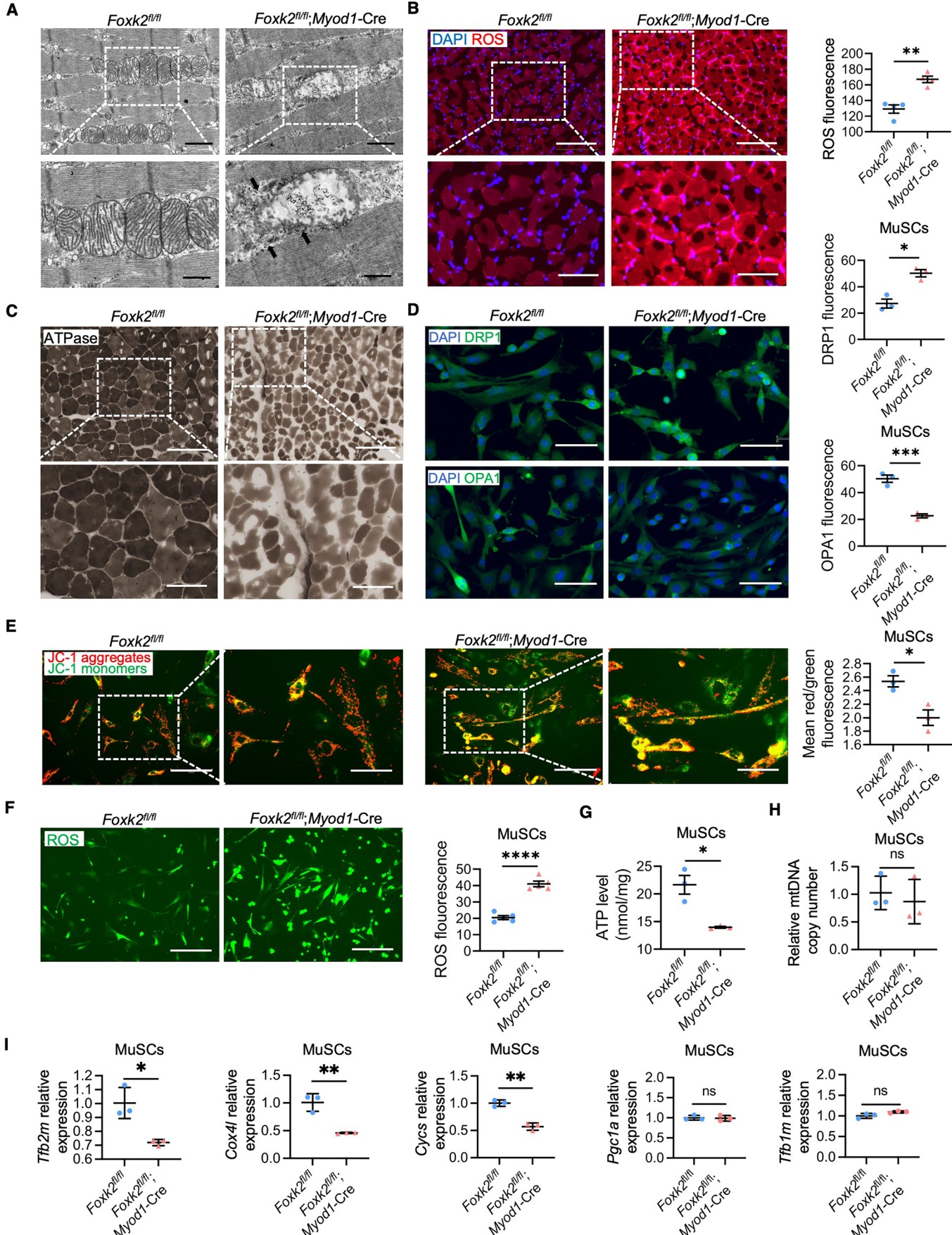

◄ **Figure 4. Defective mitochondrial functions in FOXK2 deficiency models.**

(A) Representative TEM images in TA sections of *Foxk2*^fl/fl^ and *Foxk2*^fl/fl^; *Myod1*-Cre littermates at 8 weeks. Black arrows point to the damaged mitochondria. The image below (scale bars: 1 μm) is an enlarged view of the white box in the image above (scale bars: 2 μm). (B) ROS production in TA sections of *Foxk2*^fl/fl^ and *Foxk2*^fl/fl^; *Myod1*-Cre littermates at 8 weeks by DHE staining (*n* = 4 areas). The image below (scale bars: 100 μm) is an enlarged view of the white box in the image above (scale bars: 200 μm). *P* value: **$P$ = 0.0014. (C) ATPase activity detection in TA sections of *Foxk2*^fl/fl^ and *Foxk2*^fl/fl^; *Myod1*-Cre littermates at 8 weeks. The image below (scale bars: 100 μm) is an enlarged view of the white box in the image above (scale bars: 200 μm). (D) DRP1 and OPA1 IF staining of MuSCs from *Foxk2*^fl/fl^ and *Foxk2*^fl/fl^; *Myod1*-Cre littermates with fluorescence quantitative statistics shown on the right (*n* = 3 areas). Scale bars: 100 μm. *P* value: DRP1, *$P$ = 0.0119; OPA1, ***$P$ = 0.0009. (E) Representative images of MuSCs from *Foxk2*^fl/fl^ and *Foxk2*^fl/fl^; *Myod1*-Cre littermates by JC-1 staining. The zoomed-in area (scale bars: 100 μm) is an enlarged view of the boxed area in the image (scale bars: 200 μm). The red/green ratio fluorescence quantitative statistics to measure mitochondrial membrane potential are shown on the right (*n* = 3 areas). *P* value: *$P$ = 0.0193. (F) Representative images of MuSCs (scale bars: 200 μm) and quantitative of fluorescence intensity to measure ROS levels in MuSCs from *Foxk2*^fl/fl^ and *Foxk2*^fl/fl^; *Myod1*-Cre littermates by DCFH-DA staining (*n* = 5 areas). *P* value: ****$P$ < 0.0001. (G) Quantification of the ATP content in MuSCs from *Foxk2*^fl/fl^ and *Foxk2*^fl/fl^; *Myod1*-Cre littermates (*n* = 3 repeats). The values were normalized to the total cellular protein level. *P* value: *$P$ = 0.0105. (H) Quantitative of mitochondrial DNA copy number in MuSCs from *Foxk2*^fl/fl^ and *Foxk2*^fl/fl^; *Myod1*-Cre littermates by qPCR (*n* = 3 repeats). *P* value: ns-$P$ = 0.6130. (I) Quantitative of relative mitochondrial genes expression levels in MuSCs from *Foxk2*^fl/fl^ and *Foxk2*^fl/fl^; *Myod1*-Cre littermates by qPCR (*n* = 3 repeats). *P* value: *Tfb2m*, *$P$ = 0.0125; *Cox4l*, **$P$ = 0.0039; *Cycs*, **$P$ = 0.0011; *Pgc1a*, ns-$P$ = 0.8299; *Tfb1m*, ns-$P$ = 0.0608. Data were analyzed by Student's t test. All error bars indicate mean ± standard deviation. TEM transmission electron microscopy, ROS reactive oxygen species, TA tibialis anterior, DHE dihydroethidium, IF immunofluorescence, MuSCs muscle stem cells, DCFH-DA dichlorodihydrofluorescein diacetate, qPCR quantitative real-time polymerase chain reaction. Source data are available online for this figure.

repressors (Hansen and Hodges, 2022), thereby complicating the regulatory network of gene expression. However, these observations and the specific regulatory mechanisms require further validation through future studies.

In conclusion, this study identifies FOXK2 as a novel pathogenic gene for congenital myopathy associated with ptosis and reveals its previously unreported significant role in skeletal muscle development. These findings enhance our understanding of FOXK2's contributions to both development and disease.

# Methods

## Reagents and tools table

| Reagent/Resource | Reference or Source | Identifier or Catalog Number |
|---|---|---|
| **Experimental models** | | |
| *Tg(-1.9mylpfa:EGFP)* transgenic zebrafish strain | Shanghai Model Organisms Center | *ihb16Tg/+(AB)* |
| *Foxk2*^fl/+^ mice (C57BL/6J background) | Shanghai Model Organisms Center | 6Smoc-*Foxk2*^em1(flox)Smoc^ |
| *Foxk2*^−/−^ mice (C57BL/6J background) | Shanghai Model Organisms Center | 6Smoc-*Foxk2*^em1Smoc^ |
| *Myod1*-Cre transgenic mouse model | Institute of Nutrition and Health, Chinese Academy of Sciences | kindly provided |
| C2C12 cells | Stem Cell Bank of the Chinese Academy of Sciences | CSTR:19375.09.3101MOUSCSP505 |
| Patients' samples | Eye & ENT Hospital of Fudan University | |
| **Recombinant DNA** | | |
| pCDH-FOXK2-WT-CMV-IRES-Blast | GentleGen | |
| pCDH-FOXK2-T55I-CMV-IRES-Blast | GentleGen | |
| pCDH-FOXK2-P411A-CMV-IRES-Blast | GentleGen | |
| pCDH-FOXK2-R215W-CMV-IRES-Blast | GentleGen | |
| pCDH-FOXK2-A479V-CMV-IRES-Blast | GentleGen | |

| Reagent/Resource | Reference or Source | Identifier or Catalog Number |
|---|---|---|
| pCDH-FOXK2-R544Q-CMV-IRES-Blast | GentleGen | |
| pCDH-FOXK2-Q550H-CMV-IRES-Blast | GentleGen | |
| pCDH-CRY1-CMV-IRES-Blast | GentleGen | |
| LentiCRISPRv2 | Addgene | 52961 |
| **Antibodies** | | |
| Myosin heavy chain (MyHC) | Abmart | T56635 |
| Myosin heavy chain (MyHC) | R&D systems | MAB4470 |
| FOXK2 | Abmart | PU248547 |
| FOXK2 | Novas Biologicals | NBP1-87700 |
| FOXK2 | CST | 28712T |
| Laminin beta 1 | Proteintech | 23498-1-AP |
| DPR1 | Proteintech | 12957-1-AP |
| OPA1 | Proteintech | 27733-1-AP |
| MYOG | Abcam | ab77232 |
| MYOD1 | Proteintech | 18943-1-AP |
| MYF5 | Abmart | T58504 |
| PAX7 | DSHB | AB_528428 |
| GAPDH | proteintech | 60004-1-Ig |
| FOXK1 | Abcam | ab309510 |
| PDK4 | Abmart | TD7169 |
| CRY1 | Abmart | PC11441 |
| Anti-rabbit secondary antibody | Jackson ImmunoResearch | 111-035-003 |
| YSFluor™ 488 Goat anti-rabbit secondary antibody | Yeasen | 33106ES |
| **Oligonucleotides and other sequence-based reagents** | | |
| PCR primers | This study | Table EV2 |
| RT-qPCR primers | This study | Table EV2 |
| morpholinos | This study | Table EV2 |
| **Chemicals, Enzymes and other reagents** | | |
| QIAamp DNA Blood Mini Kit | Qiagen | 51104 |
| Human All Exon V6 kit | Agilent | 5190-8863 |

| Reagent/Resource | Reference or Source | Identifier or Catalog Number |
|---|---|---|
| Hiseq PE Cluster Kit | Illumina | PE-410-1001 |
| Coenzyme Q10 | MCE | 303-98-0 |
| CoQ10 ELISA assay kit | JonInbio | JL49867 |
| Collagenase-type XI | Sigma-Aldrich | C7657 |
| Dispase II | Invitrogen | 17105041 |
| ATP assay kit | Beyotime | S0027 |
| JC-1 kit | Beyotime | C2005 |
| DCFH-DA kit | Beyotime | S0034S |
| SYBR Premix Ex Taq kit | TaKaRa | RR390Q |
| cDNA Reverse Transcription kit | TaKaRa | RR037Q |
| Chemiluminescence kit | NCM Biotech | P10060 |
| SimpleChIP® Plus Enzymatic Chromatin IP Kit | CST | 9005 |
| QiaQuick PCR purification reagents | Qiagen | 28104 |
| RNA 6000 Nano LabChip Kit | Agilent | 5067-1511 |
| MS-222 | Sigma-Aldrich | E10521 |
| Dihydroethidium | Sigma-Aldrich | D7008 |
| F-10 medium | Gibco | 11550043 |
| DMEM/high glucose media | Gibco | 11965118 |
| FBS | Gibco | A5256701 |
| Penicillin streptomycin | Gibco | 15140122 |
| Puromycin | Yeasen | 60210ES25 |
| Blasticidin | InvivoGen | Ant-bl-05 |
| Bradford Protein Assay Kit | Thermo Fisher | 23200 |
| Trizol Reagent | Invitrogen | 15596026 |
| Protein A/G magnetic beads | Thermo Fisher | 88802 |
| Tn5 Digestion Mix | TransNGS | KP101-11 |
| **Software** | | |
| Burrows-Wheeler Aligner | https://bio-bwa.sourceforge.net/ | |
| Genome Analysis Toolkit | https://gatk.broadinstitute.org/hc/en-us | |
| Cnvkit software | https://github.com/etal/cnvkit | |
| ANNOVAR | https://annovar.openbioinformatics.org/en/latest/ | |
| SnpEff | https://pcingola.github.io/SnpEff/ | |
| polymorphism phenotyping v2 | http://genetics.bwh.harvard.edu/pph2/ | |
| Mutation Taster | https://www.mutationtaster.org/ | |
| Genome Aggregation Database | http://gnomad.broadinstitute.org/ | |
| Combined Annotation Dependent Depletion | https://cadd.gs.washington.edu/ | |
| Exome Aggregation Consortium | http://exac.broadinstitute.org | |
| Chromas Lite | https://technelysium.com.au/wp/chromas/ | |
| Gene Tools | http://www.gene-tools.com/ | |
| Ethovision XT | https://noldus.com/ethovision-xt | |
| CRISPR-ERA | http://crispr-era.stanford.edu/index.Jsp | |

| Reagent/Resource | Reference or Source | Identifier or Catalog Number |
|---|---|---|
| CRISPR | http://crispr.dfci.harvard.edu/SSC/ | |
| CRISPOR | http://crispor.tefor.net/ | |
| MitoCarta3.0 | https://www.broadinstitute.org/mitocarta/mitocarta30-inventory-mammalian-mitochondrial-proteins-and-pathways | |
| Rstudio | https://posit.co/download/rstudio-desktop/ | |
| Graphpad prism | https://www.graphpad.com/ | |
| ImageJ | https://imagej.net/ij/ | |
| IGV | https://igv.org/ | |
| Chromas | https://technelysium.com.au/wp/chromas/ | |
| MEME Suite | https://meme-suite.org/meme/tools/meme | |
| Biorender | https://BioRender.com | |
| **Other** | | |
| Novaseq 6000 platform | Illumina | |

## Pedigree and sample collection

A four-generation pedigree diagnosed solely with congenital ptosis were recruited. Blood samples were collected from the proband and other seven pedigree members for WES. EM samples from the proband and a healthy individual (control) was obtained from excess discarded tissues during surgery and body donation, respectively. All tissues were preserved in 4% paraformaldehyde (PFA) for hematoxylin-eosin staining (H&E) and IHC or stored in −80 °C for qPCR experiment. Studies involving human subjects were approved by the Institutional Research Ethics Committee of Eye & ENT Hospital of Fudan University (2021016). Informed consent was obtained from all participants, and the experiments were conducted in accordance with the principles outlined in the WMA Declaration of Helsinki and Belmont Report by the U.S. Department of Health and Human Services. Permission for the publication of patient photos has been granted.

## Genetic analysis

Total DNA was extracted from blood samples using the QIAamp DNA Blood Mini Kit (Qiagen). Sequencing libraries were generated using Agilent SureSelect Human All Exon V6 kit (Agilent) following the manufacturer's protocol, with index codes added to each sample. Clustering of the index-coded samples was performed on a cBot Cluster Generation System using the Hiseq PE Cluster Kit (Illumina). After cluster generation, the DNA libraries were sequenced on Novaseq 6000 platform (Illumina), generating 150 bp paired-end reads. Valid sequencing data was mapped to the reference human genome (UCSC hg19) with the Burrows-Wheeler Aligner software. Single-nucleotide polymorphisms and small insertions and deletions in each exome were detected using the HaplotypeCaller tool of the Genome Analysis Toolkit software. Copy number variants were identified using Cnvkit. We performed variant annotation using ANNOVAR and SnpEff. Detected variants were characterized using the DbSNP and 1000 Genomes Project databases. To evaluate the conservation of mutant sites, we used polymorphism phenotyping v2 (Polyphen-2) (http://

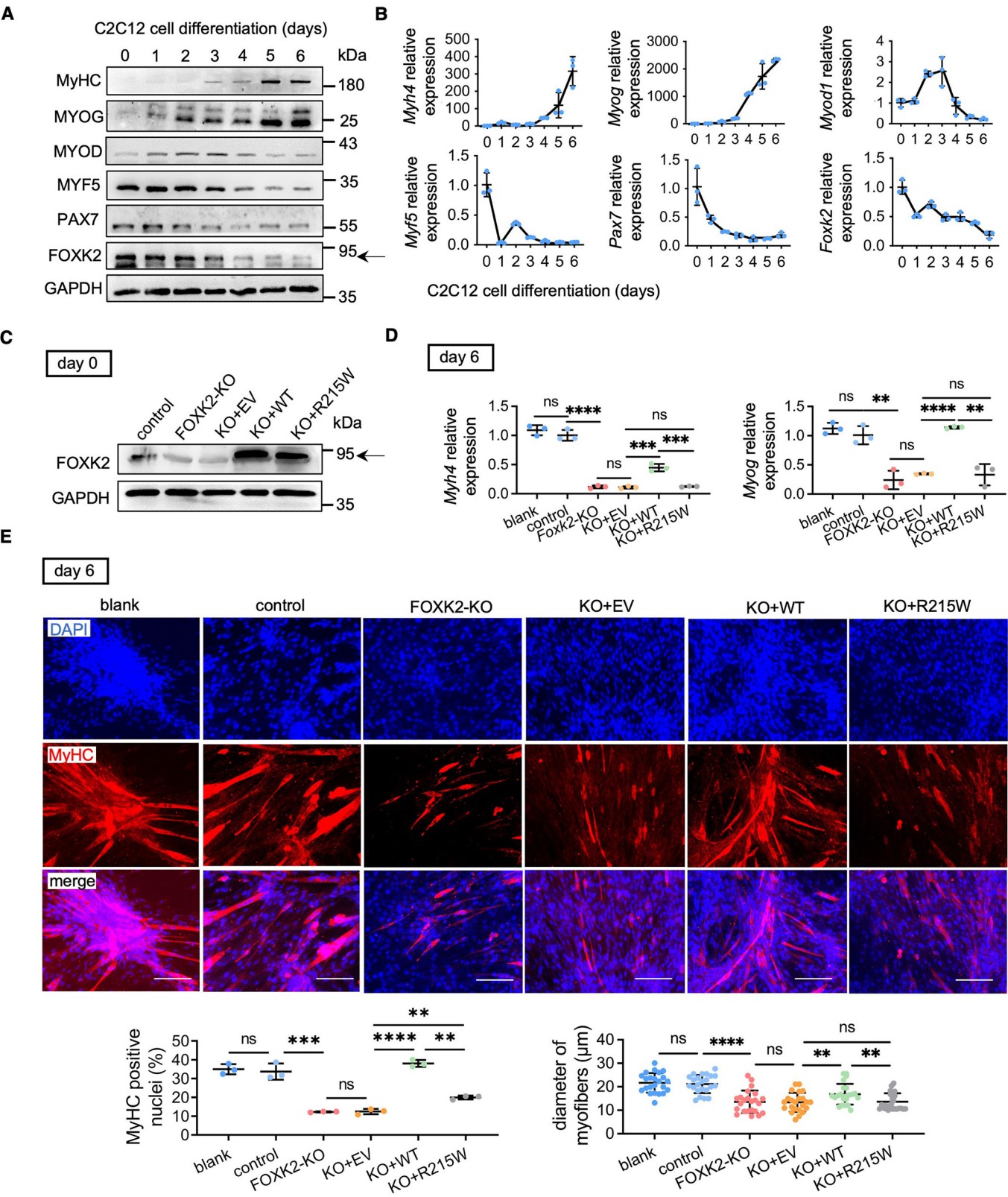

**Figure 5. Impaired myogenic differentiation by FOXK2 deficiency and ineffective rescue of R215W mutant in C2C12 cells.**

(A, B) (A)Western blot and (B) qPCR analysis ($n = 3$ repeats) of MyHC, MYOG, MYOD, MYF5, PAX7 and FOXK2 expression levels in C2C12 cells. Cells were induced into myogenic differentiation and whole cell lysates were isolated at the indicated times for analysis. The black arrow points to the specific protein band. (C) Western blot analysis of FOXK2 expression level in C2C12 cells at undifferentiated stage (day 0). The black arrow points to the specific protein band. (D) Quantitative analysis of *Myh4* and *Myog* expression levels by qPCR after 6-day myogenic differentiation ($n = 3$ repeats). P value for blank vs control, control vs FOXK2-KO, FOXK2-KO vs KO + EV, KO + EV vs KO + WT, KO + WT vs KO + R215W and KO + EV vs KO + R215W: *Myh4*: ns-$P = 0.2985$; ****$P < 0.0001$; ns-$P = 0.7651$; ***$P = 0.0009$; ***$P = 0.0009$; ns-$P = 0.2905$. *Myog*: ns-$P = 0.3348$; **$P = 0.0040$; ns-$P = 0.3182$; ****$P < 0.0001$; **$P = 0.0015$; ns-$P = 0.8951$. (E) Representative images of MyHC (red) and DAPI (blue) IF staining in C2C12 cells after 6-day myogenic differentiation (scale bars: 200 μm area). The percentage of MyHC-positive nuclei ($n = 3$ areas) and the diameter of myofibers ($n = 23$ myofibers) are shown below. P value for blank vs control, control vs FOXK2-KO, FOXK2-KO vs KO + EV, KO + EV vs KO + WT, KO + WT vs KO + R215W and KO + EV vs KO + R215W: Left: ns-$P = 0.6887$; ***$P = 0.0010$; ns-$P = 0.8348$; ****$P < 0.0001$; **$P = 0.0010$; **$P = 0.0018$. Right: ns-$P = 0.7009$; ****$P < 0.0001$; ns-$P = 0.8594$; **$P = 0.0070$; **$P = 0.0081$; ns-$P = 0.8277$. Data were analyzed by Student's t test. All error bars indicate mean ± standard deviation. Blank untreated wild-type C2C12 cells, control C2C12 cells transfected with a blank vector as a control for the FOXK2-KO group, FOXK2-KO FOXK2 knockout C2C12 cells, KO + EV FOXK2-KO cells transfected with a blank vector as a control for KO + WT and KO + R215W groups, KO + WT FOXK2-KO cells with overexpression of the human wild-type FOXK2 vector, KO + R215W FOXK2-KO cells with overexpression of the human FOXK2 R215W mutation vector, qPCR quantitative real-time polymerase chain reaction, IF immunofluorescence. Source data are available online for this figure.

genetics.bwh.harvard.edu/pph2/) and Mutation Taster (https://www.mutationtaster.org/). Predictive pathogenicity was assessed using Mutation Taster, Polyphen-2, and the Combined Annotation Dependent Depletion (CADD). Allele frequencies were explored using the Genome Aggregation Database (GnomAD) (http://gnomad.broadinstitute.org/) and the Exome Aggregation Consortium (ExAC) (http://exac.broadinstitute.org). *FOXK2* mutations were confirmed in six pedigrees by polymerase chain reaction (PCR) amplification and Sanger sequencing, and the results were visualized using Chromas Lite. All the above experiments were completed at the Medical Laboratory of Nantong Zhong Ke Co, Ltd. The WES data for pedigree 1 has been deposited in the Genome Sequence Archive (HRA008045).

## Zebrafish strains and rearing

We purchased *Tg(-1.9mylpfa:EGFP)* transgenic zebrafish strain (ihb16Tg/+(AB)) from the Shanghai Model Organisms Center. This strain exhibits green fluorescence labeling in skeletal muscle, as described in previous studies (Ma et al, 2019). Adult zebrafish were maintained at 28.5 °C on a 14-h light /10-h dark cycle. For nature mating, five to six pairs of zebrafish were set up each time. Embryos were kept at 28 °C in fish water (0.2% Instant Ocean Salt in deionized water). Embryos were washed and staged according to previously published methods (Kimmel et al, 1995). The Institutional Research Ethics Committee of Eye & ENT Hospital of Fudan University approved the study and ensured that all zebrafish procedures adhered to guidelines and recommendations outlined in the Guide for the Care and Use of Laboratory Animals.

## Zebrafish microinjections

To knock down the expression of zebrafish *foxk2*, we designed translation-blocking MOs (ATG-MO) and splice-inhibiting MOs targeting the intron 2/exon 3 splice boundary of the *foxk2* transcript (I2E3-MO) (Gene Tools, http://www.gene-tools.com/). A non-specific standard MO as a control (control-MO). We microinjected each MO 4 ng into the randomly assigned fertilized one-cell-stage embryos in each group following the standard protocol (Nasevicius and Ekker, 2000). All zebrafish embryos were microinjected simultaneously. Injected larvae were analyzed for mortality and the percentage of larvae with defect at 72-hpf, and a 72-h survival curve was established. Larvae (30–50 per group) were

randomly selected and stored in RNA-later (Beyotime) for subsequent RNA extraction. The sequences of the MOs targeting *foxk2* gene and standard MO were listed in Appendix Table S1.

## Zebrafish image acquisition

At 72-hpf, larvae were anesthetized with 0.016% MS-222 (Sigma-Aldrich). The larvae were then mounted with 3% methylcellulose in depression slides for observation by fluorescence microscopy (Nikon). Larvae were analyzed using a fluorescence microscope (Nikon) and subsequently photographed with digital cameras. Quantitative image analyses were conducted using image-based morphometric analysis and ImageJ software. The gross morphology phenotypes of 10 larvae from each treatment group were quantified analyzed.

## Zebrafish behavioral analysis

At 5-dpf, larvae from each group (control-MO, ATG-MO, and I2E3-MO) were collected, cleaned, and placed in 96-well plates. Each well contained 0.2 mL of fish water and one larva, with 10 larvae per group. Behavioral tests were performed as follows: the larvae were allowed to acclimate for 15 min before locomotion monitoring (Zhao et al, 2012). Subsequently, the larvae were allowed to freely explore the aquarium for 30 min. A camera positioned above the plate was used to track their movement. Four parameters were analyzed: total movement distance, velocity, mobility (percentage of actual moving distance to total distance), and maximum acceleration. All digital tracks were analyzed using Ethovision XT software, with a minimum movement distance of 0.2 mm to filter out system noise.

## Generation of *Foxk2* conditional knockout and global knockout mice

*Foxk2*^fl/+ mice (C57BL/6J background) and *Foxk2*^−/− mice (C57BL/6J background) were generated using CRISPR/Cas9 technology by Shanghai Model Organisms Center, Inc. In *Foxk2*^fl/+ mice, loxP sites flanking exon 2 were inserted via homologous recombination. These mice were then crossed with the *Myod1*-Cre transgenic mouse model (Kim et al, 2023a; Oprescu et al, 2023) on a C57BL/6J background to generate mice with a deletion of *Foxk2* in MuSCs. The genotyping primers are listed in Appendix Table S1. Embryos were staged with the

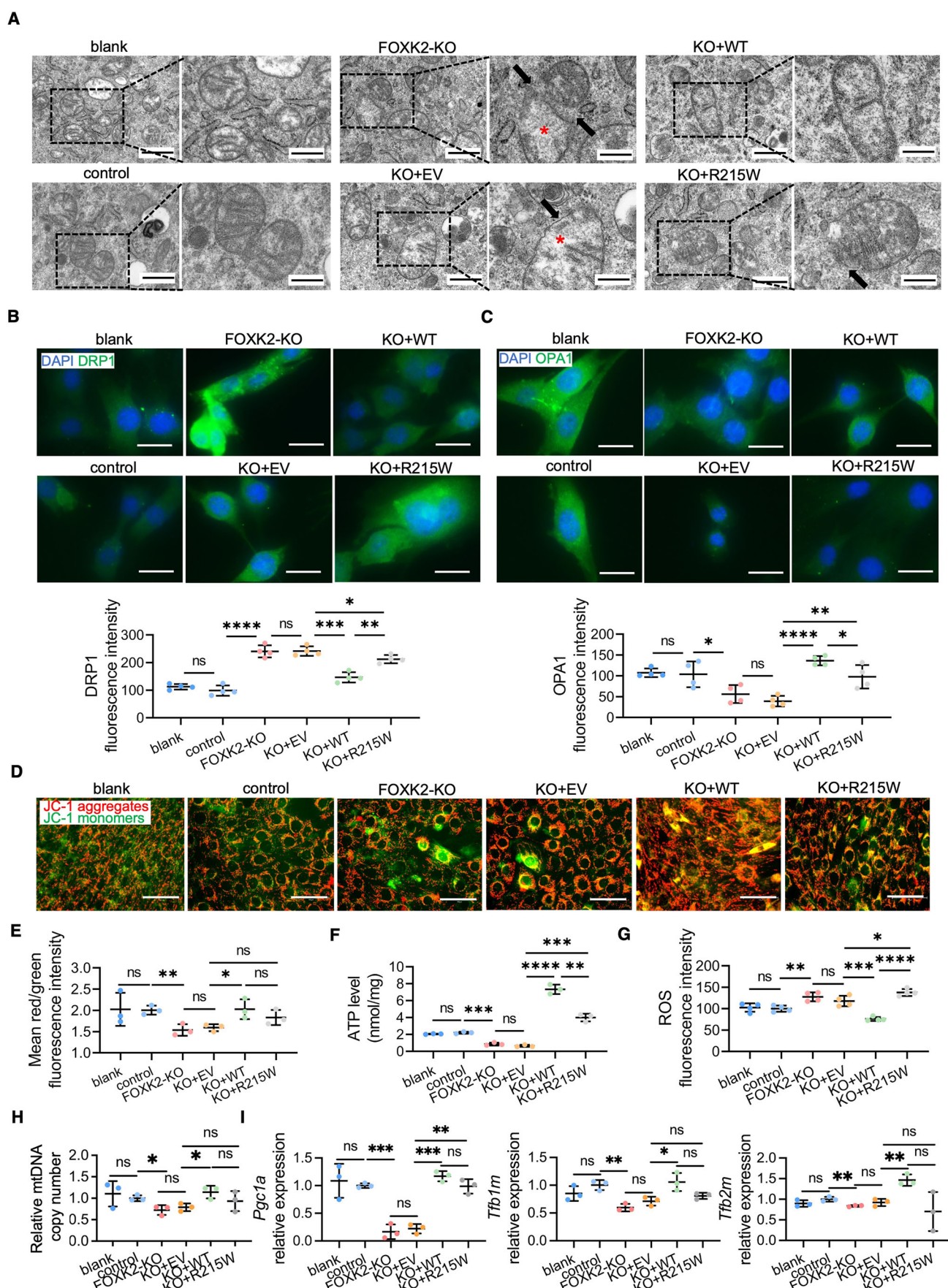

**Figure 6. Mitochondrial dyshomeostasis by FOXK2 deficiency and low-effective rescue of R215W mutant in C2C12 cells.**

(A) Representative TEM images of C2C12 cells. Blank arrows point to the incomplete mitochondrial membrane, and red asterisks indicate indistinct cristae. The zoomed-in area (scale bars: 1 μm) is an enlarged image of the boxed area in the left image (scale bars: 2 μm). (B, C) (B) DRP1 and (C) OPA1 IF staining of C2C12 cells and fluorescence quantitative statistics shown below ($n = 4$ areas). Scale bars: 50 μm. P value for blank vs control, control vs FOXK2-KO, FOXK2-KO vs KO + EV, KO + EV vs KO + WT, KO + WT vs KO + R215W and KO + EV vs KO + R215W: DRP1: ns-$P = 0.2406$; ****$P < 0.0001$; ns-$P = 0.9454$; ***$P = 0.0003$; **$P = 0.0015$; *$P = 0.0429$. OPA1: ns-$P = 0.8264$; *$P = 0.0456$; ns-$P = 0.2234$; ****$P < 0.0001$; *$P = 0.0433$; **$P = 0.0092$. (D) Representative images of C2C12 cells by JC-1 staining. Scale bars: 100 μm. (E) Quantitative statistics of mean red/green ratio fluorescence intensity of (D) to measure mitochondrial membrane potential ($n = 3$ areas). P value: **$P = 0.0097$; *$P = 0.0390$; blank vs control, ns-$P = 0.9385$; FOXK2-KO vs KO + EV, ns-$P = 0.5861$; KO + EV vs KO + R215W, ns-$P = 0.1023$; KO + WT and KO + R215W, ns-$P = 0.3174$. (F) Quantification of ATP contents in C2C12 cells ($n = 3$ repeats). The values were normalized to the total cellular protein level. P value: ****$P < 0.0001$; **$P = 0.0013$; blank vs control, ns-$P = 0.0961$; control vs FOXK2-KO, ***$P = 0.0004$; FOXK2-KO vs KO + EV, ns-$P = 0.1541$; KO + EV vs KO + R215W, ***$P = 0.0003$. (G) Quantitative analysis of fluorescence intensity to measure ROS levels in C2C12 cells by DCFH-DA staining ($n = 4$ repeats). P value: ****$P < 0.0001$; ***$P = 0.0007$; **$P = 0.0050$; *$P = 0.0337$; blank vs control, ns-$P = 0.7027$; FOXK2-KO vs KO + EV, ns-$P = 0.2653$. (H) Quantitative of mtDNA copy number in C2C12 cells measured by qPCR ($n = 3$ repeats). P value: blank vs control, ns-$P = 0.5916$; control vs FOXK2-KO, *$P = 0.0214$; FOXK2-KO vs KO + EV, ns-$P = 0.4779$; KO + EV vs KO + WT, *$P = 0.0234$; KO + WT vs KO + R215W, ns-$P = 0.2577$; KO + EV vs KO + R215W, ns-$P = 0.3716$. (I) Quantitative of relative mitochondrial relative gene expression levels in C2C12 cells by qPCR ($n = 3$ repeats). P value for blank vs control, control vs FOXK2-KO, FOXK2-KO vs KO + EV, KO + EV vs KO + WT, KO + WT vs KO + R215W and KO + EV vs KO + R215W: *Pgc1a*: ns-$P = 0.6530$; ***$P = 0.0005$; ns-$P = 0.5826$; ***$P = 0.0002$; ns-$P = 0.1210$; **$P = 0.0011$. *Tfb1m*: ns-$P = 0.1746$; **$P = 0.0032$; ns-$P = 0.1248$; *$P = 0.0302$; ns-$P = 0.0663$; ns-$P = 0.1662$. *Tfb2m*: ns-$P = 0.1434$; **$P = 0.0064$; ns-$P = 0.4715$; **$P = 0.0047$; ns-$P = 0.0556$; ns-$P = 0.4704$. Data were analyzed by Student's t test. All error bars indicate mean ± standard deviation. Blank untreated wild-type C2C12 cells, control C2C12 cells transfected with a blank vector as a control for the FOXK2-KO group, FOXK2-KO FOXK2 knockout C2C12 cells, KO + EV FOXK2-KO cells transfected with a blank vector as a control for KO + WT and KO + R215W groups, KO + WT FOXK2-KO cells with overexpression of the human wild-type FOXK2 vector, KO + R215W FOXK2-KO cells with overexpression of the human FOXK2 R215W mutation vector, ROS reactive oxygen species, TEM transmission electron microscopy, IF immunofluorescence, DCFH-DA Dichlorodihydrofluorescein diacetate, mtDNA mitochondrial DNA, qPCR quantitative real-time polymerase chain reaction. Source data are available online for this figure.

morning of the vaginal plug considered as E0.5. All transgenic mice were backcrossed to wild-type mice (C57BL/6J background) after every three generations of mating. The efficiency of *Foxk2* knockout was confirmed via IHC staining. Eight-week-old male mice were selected for body weight measurements, Gas weight/body weight ratio statistics, and behavioral experiments. Their skeletal muscle tissues were collected for histological examination. Additionally, skeletal muscle tissues from male mice at postanal days 1, 4 weeks and 8 weeks were selected for H&E staining. All experiments were conducted with the approval of the Experimental Animal Ethics Committee of the Eye & ENT Hospital of Fudan University.

## Coenzyme Q10 supplementation

In the mouse study, we selected an optimal administration method for CoQ10 (MCE) supplementation, delivering 600 mg/kg/day (dissolved in corn oil) via oral gavage, as previously described (Pan et al, 2024). *Foxk2*^fl/fl^; *Myod1*-Cre mice, aged at 2 weeks and exhibiting skeletal muscle dysplasia, were randomly assigned to receive either freshly prepared CoQ10 or a vehicle control for 3 days before being euthanized on the 7th day. We chose the TA muscle to evaluate CoQ10 content using a CoQ10 ELISA assay kit (Jonlnbio), following the manufacturer's instructions.

## Hanging endurance test

An inverted grid suspension experiment was conducted to assess the gripping power of the mouse limbs. Each animal was placed in the center of a 21 cm × 21 cm wire grid (line width of approximately 0.1 cm, spacing, 0.5 cm). The grid was gently tapped to encourage the mouse to grip tightly, after which the grid was slowly inverted to a horizontal position. The duration time for which the mouse remained hanging onto the grid was recorded. Animals underwent training twice daily for 3 consecutive days, with a 180-s cutoff time. The experiment was repeated three times per mouse (5–6 mice), and the average hanging time value for each mouse was calculated as the evaluation metric.

## Hematoxylin-eosin staining

H&E staining on paraffin sections were performed using a standard protocol. Six-mm-thick section slides were first stained in hematoxylin for 10 min, followed by thorough rinsing under running tap water for at least 3 min. The slides were then immersed in 0.2% acid alcohol for 1 s and immediately rinsed under running tap water. Next, the slides were stained in eosin for 2 min, followed by rinsing and dehydrating in graded ethanol and Xylene. Finally, the 4 fields of view of slides were randomly selected to observe under a light microscope (Invitrogen) and statistically counted.

## Immunohistochemistry

Tissues preserved in 4% PFA were embedded in paraffin and sectioned. After hydration through a graded ethanol series (100% to 50%), the sections were incubated in 3% $H_2O_2$ for 10 min to quench endogenous peroxidase activity. The sections were then blocked with goat serum (Sigma-Aldrich) before being incubated with the primary antibody overnight at 4 °C. The following day, the sections were incubated with a secondary antibody and processed using an ultrasensitive DAB kit (Sigma-Aldrich). Four images were randomly selected to capture using a light microscope (Invitrogen). Quantitative analysis of positive brown cell was performed on three or four fields per sample, with equal magnification, using ImageJ software. The antibodies used are listed in Appendix Table S2.

## Immunofluorescence

For mouse samples, muscle tissues were isolated and fixed in 4% PFA. Successive 8 μm sections were cut for further experiments. PFA was removed using xylene, and samples were dehydrated with ethanol of graded concentrations. Antigen retrieval was performed using high-pressure cooking in 1 mM EDTA buffer (pH = 9.0). Tissue sections were blocked with 5% bovine serum albumin and 15% goat serum in 1× phosphate buffer saline (PBS) for 1 h. Primary antibodies were incubated overnight 4 °C, followed by

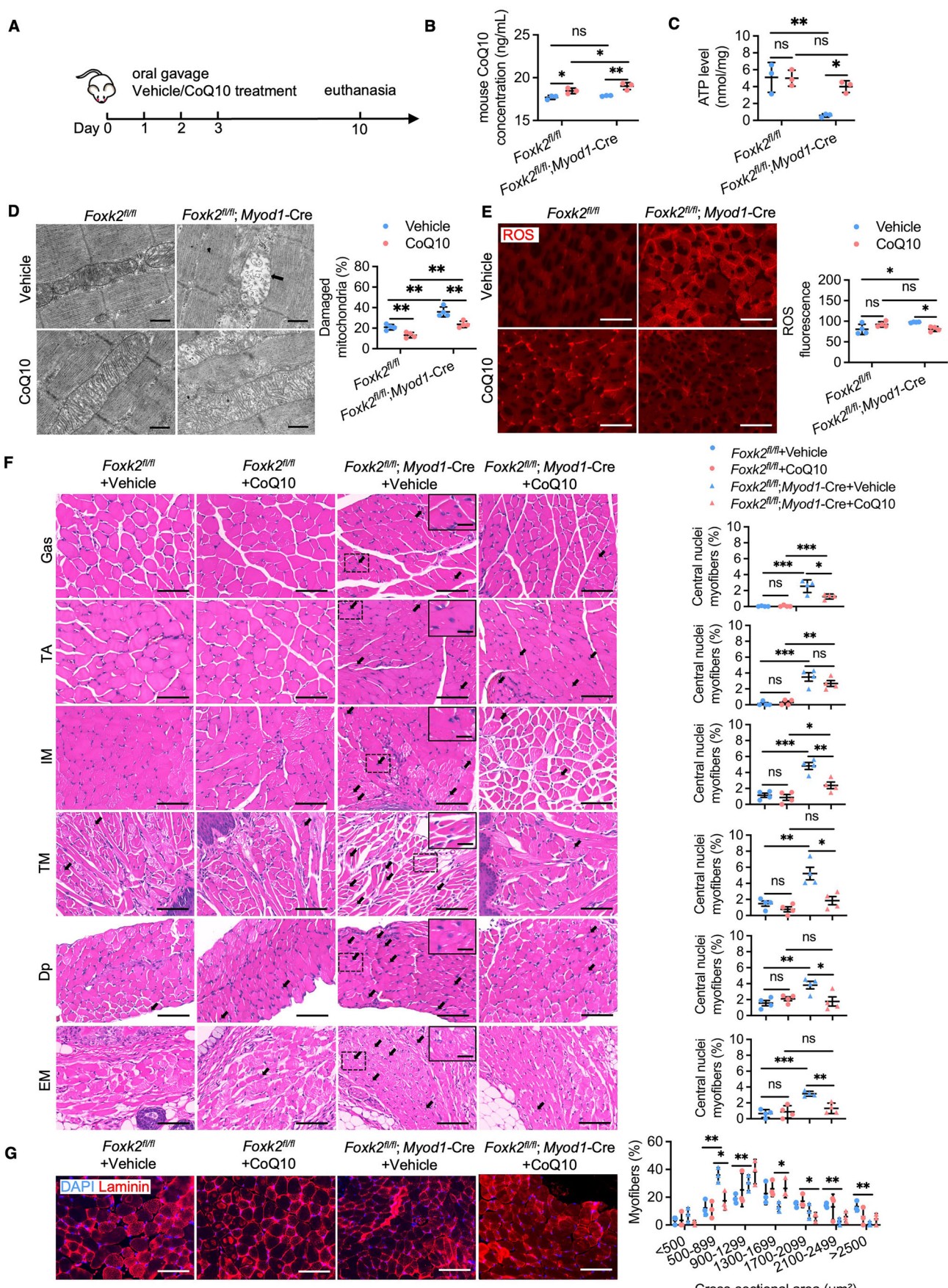

**Figure 7. Coenzyme Q10 improves mitochondrial dysfunction and skeletal muscle development disorders of mice with *Foxk2* deficiency in muscle stem cells.**

(A) Illustration of the study design of CoQ10 treatment in mice. (B) Concentration of CoQ10 in TA tissues of mice at 2 weeks ($n = 3$ mice). P value: **$P = 0.0013$; ns-$P = 0.4524$; $Foxk2^{fl/fl}$+Vehicle vs $Foxk2^{fl/fl}$ + CoQ10, *$P = 0.0118$; $Foxk2^{fl/fl}$ + CoQ10 vs $Foxk2^{fl/fl}$;$Myod1$-Cre+CoQ10, *$P = 0.0435$. (C) Quantification of ATP content in TA tissues from $Foxk2^{fl/fl}$ and $Foxk2^{fl/fl}$; $Myod1$-Cre littermates at 2 weeks, with or without CoQ10 treatment ($n = 3$ mice). Values were normalized to total protein levels. P value: **$P = 0.0012$; *$P = 0.0119$; $Foxk2^{fl/fl}$+Vehicle vs $Foxk2^{fl/fl}$ + CoQ10, ns-$P = 0.9377$; $Foxk2^{fl/fl}$ + CoQ10 vs $Foxk2^{fl/fl}$;$Myod1$-Cre+CoQ10, ns-$P = 0.2180$. (D) Representative TEM images of TA sections from $Foxk2^{fl/fl}$ and $Foxk2^{fl/fl}$; $Myod1$-Cre littermates at 2 weeks. Black arrows indicate damaged mitochondria. The percentage of damaged mitochondria was quantified using ImageJ software ($n = 8$ areas). Scale bars: 1 μm. P value: $Foxk2^{fl/fl}$+Vehicle vs $Foxk2^{fl/fl}$;$Myod1$-Cre+Vehicle **$P = 0.0017$; $Foxk2^{fl/fl}$+Vehicle vs $Foxk2^{fl/fl}$ + CoQ10, **$P = 0.0066$; $Foxk2^{fl/fl}$ + CoQ10 vs $Foxk2^{fl/fl}$;$Myod1$-Cre+CoQ10 **$P = 0.0022$; $Foxk2^{fl/fl}$;$Myod1$-Cre+Vehicle vs $Foxk2^{fl/fl}$;$Myod1$-Cre+ CoQ10, **$P = 0.0057$. (E) ROS production in TA sections of $Foxk2^{fl/fl}$ and $Foxk2^{fl/fl}$; $Myod1$-Cre littermates at 2 weeks, with or without CoQ10 treatment, measured by DHE staining ($n = 4$ areas). Quantifications were performed using ImageJ software. Scare bar: 200 μm. P value: $Foxk2^{fl/fl}$+Vehicle vs $Foxk2^{fl/fl}$;$Myod1$-Cre+Vehicle *$P = 0.0325$; $Foxk2^{fl/fl}$+Vehicle vs $Foxk2^{fl/fl}$ + CoQ10, ns-$P = 0.1736$; $Foxk2^{fl/fl}$ + CoQ10 vs $Foxk2^{fl/fl}$;$Myod1$-Cre+CoQ10 ns-$P = 0.2200$; $Foxk2^{fl/fl}$;$Myod1$-Cre+Vehicle vs $Foxk2^{fl/fl}$;$Myod1$-Cre + CoQ10, *$P = 0.0426$. (F) H&E staining illustrating histological aspects and the percentage of central nuclei myofibers in skeletal muscles (Gas, TA, IM, TM, Dp, and EM) of $Foxk2^{fl/fl}$ and $Foxk2^{fl/fl}$; $Myod1$-Cre littermates at 2 weeks, with or without CoQ10 treatment ($n = 4$ areas). Black arrows indicate central nuclei. The inset in the upper right corner (scale bars: 25 μm) is an enlarged view of the boxed area in the main image, highlighting central nuclei myofibers (scale bars: 100 μm). P value for $Foxk2^{fl/fl}$+Vehicle vs $Foxk2^{fl/fl}$ + CoQ10, $Foxk2^{fl/fl}$;$Myod1$-Cre+Vehicle vs $Foxk2^{fl/fl}$;$Myod1$-Cre+CoQ10, $Foxk2^{fl/fl}$+Vehicle vs $Foxk2^{fl/fl}$;$Myod1$-Cre+Vehicle and $Foxk2^{fl/fl}$ + CoQ10 vs $Foxk2^{fl/fl}$;$Myod1$-Cre+CoQ10: Gas: ns-$P = 0.6529$; *$P = 0.0225$; ***$P = 0.0003$; ***$P = 0.0007$. TA: ns-$P = 0.4640$; ns-$P = 0.2498$; ***$P = 0.0008$; **$P = 0.0011$. IM: ns-$P = 0.5588$; **$P = 0.0067$; ***$P = 0.0004$; *$P = 0.0337$. TM: ns-$P = 0.1669$; *$P = 0.0110$; **$P = 0.0044$; ns-$P = 0.1219$. Dp: ns-$P = 0.2431$; *$P = 0.0322$; **$P = 0.0077$; ns-$P = 0.6149$. EM: ns-$P = 0.6625$; **$P = 0.0024$; ***$P = 0.0001$; ns-$P = 0.4264$. (G) IF staining of Laminin in the TA of $Foxk2^{fl/fl}$ and $Foxk2^{fl/fl}$; $Myod1$-Cre littermates at 2 weeks with or without CoQ10 treatment. The distribution of CSA of myofibers was calculated ($n = 3$ areas). Scale bars: 100 μm. P value: 500-899, **$P = 0.0063$, *$P = 0.0198$; 900-1299, **$P = 0.0010$; 1300-1699, *$P = 0.03871$; 1700-2099, *$P = 0.0251$; 2100-2499, **$P = 0.0022$; >2500, **$P = 0.0060$. Data were analyzed by Student's t test. All error bars indicate mean ± standard deviation. CoQ10 Coenzyme Q10, Gas gastrocnemius, TA tibialis anterior, IM intercostal muscle, TM tongue muscle, Dp diaphragm, EM eyelid muscle, TEM transmission electron microscopy, ROS reactive oxygen species, DHE dihydroethidium, H&E hematoxylin-eosin staining, IF immunofluorescence, CSA cross-sectional area.

incubation with Alexa Fluor-conjugated secondary antibodies at room temperature for 2 h. For cellular immunostaining, cells were fixed in 4% PFA for 15 min, permeabilized with 0.2% Triton X-100 in PBS, and blocked with 5% goat serum. This was followed by overnight incubation with primary antibodies at 4 °C and subsequent incubation with secondary antibodies and DAPI for 1 h. Three or four immunofluorescence (IF) images were randomly selected using a fluorescence microscope (Invitrogen). Quantifications were performed using ImageJ software. The antibodies used are listed in Appendix Table S2.

## Mitochondrial morphology assays

In TEM assay, tissues and cell samples were washed with PBS and then fixed in 2.5% glutaraldehyde and 0.1 M sodium cacodylate buffer at 4 °C overnight. The samples were then embedded and subjected to ultramicrotomy. TEM images were captured using a microscope (Japan Electron Optics Laboratory) and analyzed with iTEM 5.2 software. Quantifications were performed using ImageJ software. In confocal microscope assay, adherent cells were incubated in serum-free medium with mito-tracker probe at 37 °C for 30 min. After aspirating the staining solution, the cells were washed and soaked in fresh medium before being observed under a confocal microscope (Leica). Quantifications were also performed using ImageJ software.

## Reactive oxygen species level measurement

For mouse samples, muscle tissues were isolated and embedded. Successive 8 μm sections were cut using freezing microtome (Thermo Fisher). These sections were stained with 1 μM dihydroethidium (Sigma-Aldrich) to evaluate ROS levels. Four images were randomly captured using a fluorescence microscope (Leica). For cell samples, cells were incubated with DCFH-DA (Beyotime) following the manufacturer's instructions. Four or five images were acquired using a fluorescence microscope (Invitrogen). Quantifications were performed using ImageJ software.

## ATPase staining

All sections were preincubation in a 0.04 M calcium solution (pH 9.3) for 15 min. Some serial sections underwent additional pre-incubation in an acid solution (pH 4.5 or pH 4.3) for 5 min. Subsequently, all sections were then incubated for 10 min at 37 °C with 18 mM $CaCl_2$ and 4 mM ATP (Sigma-Aldrich) (pH 9.4). Images were captured using a light microscope (Invitrogen).

## Primary muscle stem cells extraction

Primary MuSCs were isolated from the hindlimbs of postanal days 7 male wild-type mice, as described in previous study (Guo et al, 2022). Muscles were dissected, minced, and dissociated in 0.2% (wt/vol) collagenase-type XI (Sigma-Aldrich) and 2.4 U/mL dispase II (Invitrogen) in DMEM (Gibco) at 37 °C. Following centrifugation, the pellet was filtered through an 80 μm cell strainer and resuspended in growth medium containing F-10 (Gibco) supplemented with 20% fetal bovine serum (FBS) (Gibco) and 1% penicillin/streptomycin (Gibco). Cells were then laid repeatedly on the dish for 3 h to remove adherent cells. The unattached cells (MuSCs) were harvested and plated on dishes coated with the Matrigel (1% v/v Matrigel in DMEM) (Gibco). Cells were not allowed to reach more than 75% confluence during passages. Primary myoblasts were incubated in growth medium containing F-10 supplemented with 20% FBS and 1% penicillin/streptomycin (Gibco).

## Measurement of mitochondrial membrane potential

Cells were seeded at a density of $2 \times 10^5$ cells/well in a 12-well plate and cultured for 24 h. Trypsinized cells were stained with 2 μM 5,5',6,6'-tetrachloro-1,1',3,3'-tetraethyl-benzimidazolylcarbocyanine chloride (JC-1) (Beyotime) for 30 min. Three images were captured by fluorescence microscope (Invitrogen) and the red (aggregates) and green (monomers) fluorescence intensity were analyzed using imageJ software.

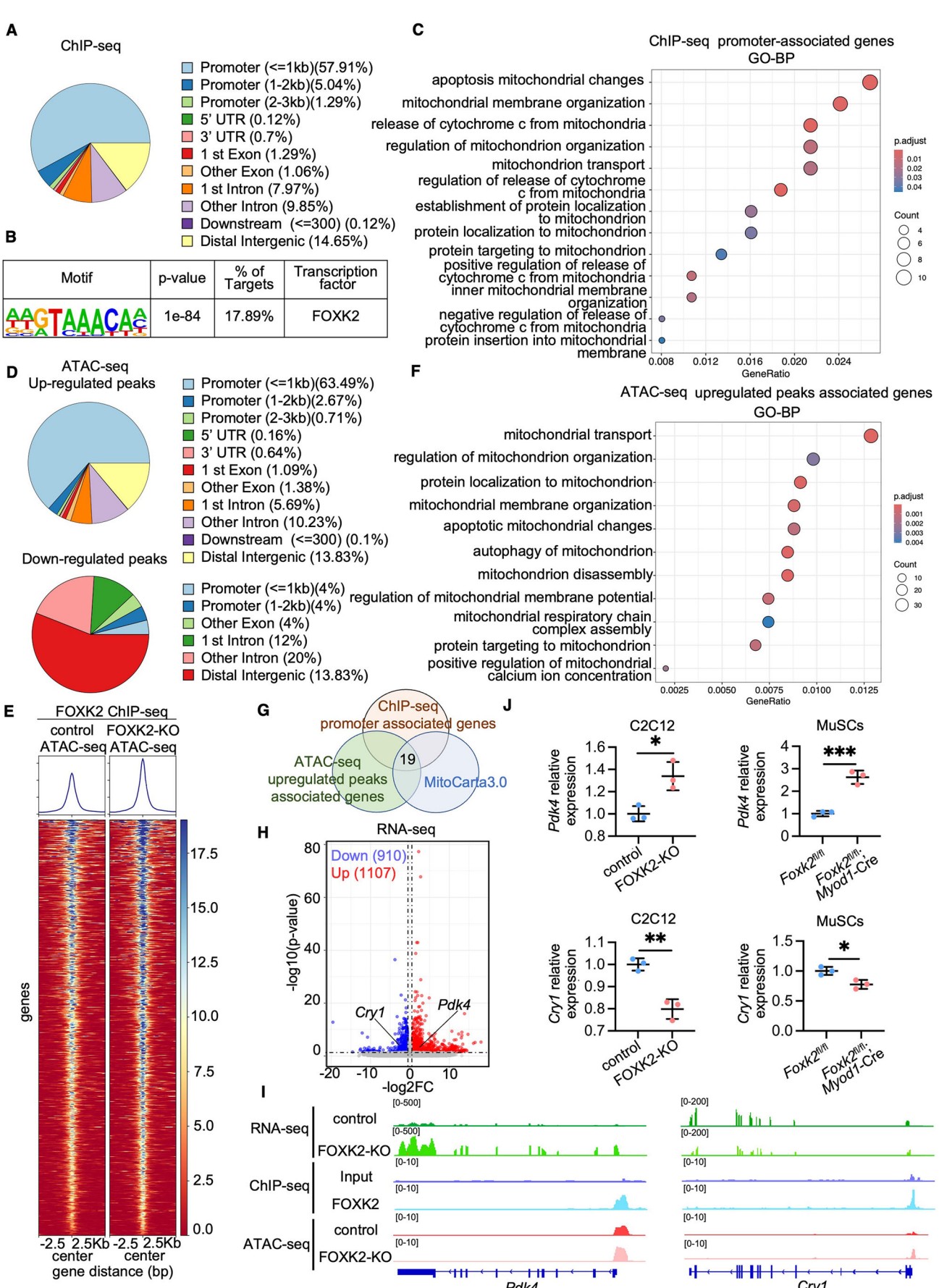

**Figure 8. FOXK2 transcriptional regulation in C2C12 Cells.**

(A) Genome-wide distribution of FOXK2 binding sites identified by ChIP-seq in C2C12 cells. (B) Enriched motif at FOXK2 binding sites identified by ChIP-seq. (C) GO analysis (BP) of genes with promoter regions bound by FOXK2, with a focus on mitochondrion-related terms. (D) Genome-wide distribution of FOXK2-binding peaks with altered chromatin accessibility in FOXK2-KO C2C12 cells, as determined by ATAC-seq. (E) Heatmap showing changes in chromatin accessibility changes at FOXK2 binding sites in FOXK2-KO C2C12 cells. (F) GO analysis (BP) of genes with upregulated chromatin accessibility in FOXK2-KO C2C12 cells, highlighting mitochondrion-related terms. (G) Venn diagrams show the overlapping (19 genes) among genes FOXK2-bound promoters (ChIP-seq), upregulated chromatin accessibility (ATAC-seq), and mitochondrion-related genes from the MitoCarta 3.0 database. (H) Volcano plot of DEGs (Down: 910 genes; Up: 1107 genes) from RNA-seq, with genes showing |log2FC| > 0.58 and p-value < 0.05. (I) Normalized RNA-seq, ChIP-seq and ATAC-seq profiles at the *Pdk4* and *Cry1* loci in FOXK2-KO *vs.* control cells. (J) Quantitative of relative *Pdk4* and *Cry1* mRNA expression levels in C2C12 cells and MuSCs from *Foxk2*^fl/fl^ and *Foxk2*^fl/fl^; *Myod1*-Cre mice, assessed by qPCR (n = 3 repeats). P value: C2C12: *Pdk4*, *P = 0.0154; *Cry1*, **P = 0.0027. MuSCs: *Pdk4*, ***P = 0.0010; *Cry1*, *P = 0.0176. Data were analyzed by Student's t test. All error bars represent mean ± standard deviation. MuSCs muscle stem cells, ChIP-seq chromatin immunoprecipitation sequencing, GO Gene Ontology, BP biological process, control C2C12 cells transfected with a blank vector as a control for the FOXK2-KO group, FOXK2-KO FOXK2 knockout C2C12 cells, ATAC-seq assay of transposase accessible chromatin sequencing, RNA-seq RNA sequencing, DEGs differentially expressed genes, FC fold change, qPCR quantitative real-time polymerase chain reaction. Source data are available online for this figure.

## ATP measurement

ATP concentrations in cells or tissues were determined luminometrically using ATP assay kits (Beyotime) following the manufacturer's instructions. The ATP concentration and standard curve were established by a luminometer (PerkinElmer). ATP levels was normalized to the protein content of each sample, which was measured using a Bradford Protein Assay Kit (Thermo Fisher). Data are representative of three independent experiments and were analyzed.

## Quantitative real-time polymerase chain reaction

Total RNA was extracted from animal tissues or cultured cells using Trizol Reagent (Invitrogen). The cDNA was synthesized with the cDNA Reverse Transcription kit (Takara). Gene expression was assessed using standard qPCR methods with the SYBR Premix Ex Taq kit (TaKaRa). Analysis was performed on a Real-Time PCR System (Thermo Fisher). Data are representative of three independent experiments and were analyzed. The primer sequences used are listed in Appendix Table S1.

## Mitochondrial DNA copy number quantification

The ratio of the mtDNA-encoded *Mt-Co2* to nuclear encoded *Gapdh* were evaluated using the $2^{\Delta\Delta Ct}$ method. Data are representative of three independent experiments and were analyzed. All primers are listed in Appendix Table S1.

## Cell culture

C2C12 cells and HEK-293T cells were obtained from the Cell Bank/Stem Cell Bank of the Chinese Academy of Sciences and cultured at 37 °C in growth medium containing DMEM (Gibco) supplemented with 10% FBS (Gibco) and 1% penicillin/streptomycin (Gibco). Myogenic differentiation of C2C12 cells was induced by replacing the growth medium with differentiation medium (DMEM supplemented with 2% horse serum (Gibco) and 1% penicillin/streptomycin (Gibco)).

## Plasmid construction

The sgRNA for CRISPR/Cas9 were designed using CRISPR-ERA (http://crispr-era.stanford.edu/index.Jsp), CRISPR (http://crispr.dfci.harvard.edu/

SSC/), and CRISPOR (http://crispor.tefor.net/) tools, with a focus on sequences within 100 amino acids downstream of the start codon, ensuring high score, good specificity, and low off-target rates. The selected sgRNA sequences were then recombined with the CRISPR/Cas9 plasmid (lentiCRISPR v2-puro) and transfected into C2C12 cells. Gene overexpression plasmids were constructed using the pCDH-CMV-IRES-blast vector and then transfected into C2C12 or HEK-293T cells. Following transfection, cells were subjected to puromycin or blasticidin screening for 2 days to isolate successfully transfected cells. Knockout or overexpression efficiency was subsequently assessed using western blot.

## Western blot

Cells were lysed in RIPA lysis buffer (NCM Biotech), supplemented with a complete protease inhibitor cocktail (Beyotime). Equal amounts of protein were loaded onto 10–12.5% SDS-PAGE gels and subsequently transferred to nitrocellulose membranes (Pall). Following transfer, the membranes were blocked with 8% milk dissolved in Tris-buffered saline with Tween (TBST) for 1 h at room temperature. Subsequently, the membranes were incubated with primary antibodies overnight at 4 °C, followed by incubation with HRP-conjugated secondary antibodies for 1 h. Specific signals were visualized using a chemiluminescence kit (NCM Biotech). The antibodies used are listed in Appendix Table S2.

## Chromatin immunoprecipitation sequencing

C2C12 cells ($5$–$10.0 \times 10^6$) were crosslinked with 1% formaldehyde and then quenched in a final solution at a final concentration of 0.125 M. The chromatin was extracted and digested with micrococcal nuclease to primarily generate mononucleosomes, with a minor fraction of dinucleosomes. Digestion reaction was halted by adding 8 μL of a solution containing 0.2 M EDTA and 0.2 M EGTA. The chromatin was then incubated overnight at 4 °C with either 2 μL of rabbit anti-FOXK2 antibody or 2 μL control rabbit IgG antibody. Protein A/G magnetic beads (Thermo Fisher) were added for an additional 4 h incubation. The resulting immunoprecipitates were washed thoroughly to remove unbound materials, and DNA was purified according to the manufacturer's instructions using the SimpleChIP® Plus Enzymatic Chromatin IP Kit (CST). For ChIP-seq, the ChIP-enriched DNA was subjected to end-repair, dA-tailing, and linker ligation, followed by barcoding and PCR amplification to add Illumina adapters. Libraries were purified

**The paper explained**

**Problem**

Congenital ptosis, a condition primarily caused by the under-development or dysfunction of the levator palpebrae muscle, affects both vision and appearance. Genetic factors are key in its pathogenesis. This disorder is often part of a broader spectrum of phenotypic mani-festations observed in severe congenital myopathies. Mutations in genes linked to congenital ptosis may also lead to more widespread skeletal muscle development disorders. Notably, ptosis is often one of the earliest signs of congenital myopathies, serving as an early indicator of more extensive muscular involvement. Despite its known genetic basis, research into the genetic mechanisms underlying congenital ptosis remains limited.

**Results**

In this study, we identified *FOXK2* mutations with potential patho-genicity in five pedigrees with congenital myopathy associated with ptosis. In zebrafish models, *Foxk2* deficiency led to underdeveloped skeletal muscles and reduced mobility. In mice with *Foxk2* deletion in skeletal muscle stem cells (MuSCs), we observed generalized skeletal muscle abnormalities and impaired mitochondrial function in MuSCs, which could be partially rescued by coenzyme Q10 supplementation. In C2C12 myoblast cell lines, loss of FOXK2 impaired myogenic differ-entiation and mitochondrial function. Rescue experiments further con-firmed the loss-of-function effect of FOXK2 mutation. Preliminary omics analysis suggested that FOXK2 regulates mitochondrial genes by modulating chromatin accessibility at its binding sites.

**Impact**

Our findings establish *FOXK2* as a novel pathogenic gene in congenital myopathy associated with ptosis, highlighting its previously unrecog-nized role in skeletal muscle development. These results expand our understanding of FOXK2's role in both normal development and dis-ease, and suggest potential therapeutic strategies for congenital myo-pathy associated with ptosis caused by *FOXK2* mutations.

using QiaQuick PCR purification reagents (Qiagen) and size-selected by 0.7× and 0.2× Ampure XP beads (Beckman). All experiments were conducted with three biological replicates. The antibodies used are listed in Appendix Table S2.

## Analysis of chromatin immunoprecipitation sequencing data

Adapter sequences and low-quality reads were removed from the sequencing data using Fastp (v.0.20.0) with the following parameters: "-thread 8 $-5$ $-3$ -W 4." The paired-end reads were then aligned to the reference genome using Bowtie2 (v.2.3.5.1) with the parameter "-p 8 –sensitive" setting. Duplicated reads were removed using Picard (v.2.22.8) with the option "REMOVE_DU-PLICATES=true." The fragment ratio in peaks (FRIP) was calculated using Bedtools (v.2.29.2) and awk (v.4.0.2). For data visualization, a bigWig file was generated using deepTools to normalize counts per million (CPM), and the data were visualized in Integrative Genomics Viewer (IGV). Peak calling was performed using HOMER, with the "findPeaks" program and the "-style histone" option to identify enriched regions. These enriched peak regions were then input for DESeq2 (v.1.30.0) to identify differential peaks. Additionally, motif analysis was conducted using the MEME Suite (https://meme-suite.org/meme/tools/meme).

## Assay of transposase accessible chromatin sequencing

To profile open chromatin regions, a total of 50,000 control and FOXK2 knockout C2C12 cells were washed once with cold PBS. Cells were lysed on ice for 3 min in 50 μL of ice-cold lysis buffer containing 10 mM Tris (pH 7.4), 10 mM NaCl, 3 mM MgCl$_2$, 0.1% NP-40, 0.1% Tween 20, and 0.01% Digitonin in diethylpyrocarbonate (DEPC)-treated water. Following lysis, cells were resuspended in ice-cold RBS-Wash buffer (10 mM Tris, pH 7.4, 10 mM NaCl, 3 mM MgCl$_2$, 0.1% Tween 20) and pelleted at 4 °C at $500 \times g$ for 10 min. Tagmentation was performed in 1× Tagmentation buffer composed of 10 mM Tris (pH 7.4), 5 mM MgCl$_2$, 10% DMF, 33% PBS, 0.1% Tween 20, and 0.01% Digitonin, using 100 nM Tn5 Transposase for 15 min at 37 °C. An equal volume of 2x Tn5 Digestion Mix (TransNGS) was added, and the tagmentated samples were purified using the QIAquick PCR Purification Kit (Qiagen). Libraries were then amplified with Illumina primers, purified again, and size-selected using 0.7× and 0.2× DNA clear beads (TransNGS). Finally, ATAC-seq libraries were sequenced on a NovaSeq platform (Anoroad) using paired-end reads of 150 base pairs (bp). All experiments were conducted with three biological replicates.

## Analysis of transposase accessible chromatin sequencing data

Adapter sequences and low-quality reads were removed from the sequencing data using Fastp (v.0.20.0). The remaining reads were aligned to the mouse genome (GRCm38/mm10) using Bowtie2 (v.2.3.5.1) with the parameters "-sensitive, -X 2000." Duplicate reads, which arise during PCR amplification, were removed using Picard (v2.22.8). The fragment ratio in peaks (FRIP) was calculated with Bedtools (v.2.29.2) and awk (v.4.0.2). To visualize the data, a bigWig file was generated using deepTools (v.3.5.1) with reads per kilobase per million normalization, and the data were visualized in IGV. SAM files were converted to BAM format using Samtools (v.0.1.19), and peak calling was performed with MACS2(v.2.2.4) using the parameters "-t input_file -q 0.01 -f BAM -nomodel -shift $-100$ -extsize 200 -keep-dup all". Differentially enriched regions between experimental and control groups were identified using DiffBind (v.2.14.0). Differentially enriched genes were analyzed using edgeR and DESeq2.

## RNA sequencing and data analysis

Total RNA was extracted from C2C12 cells using TRIzol reagent (Invitrogen), and the RNA quantity and purity were assessed using the Bioanalyzer 2100 and RNA 6000 Nano LabChip Kit (Agilent). Only samples with an RNA integrity number (RIN) > 7.0 were selected for subsequent library construction. Paired-end (PE) RNA-seq was performed on the Illumina Novaseq™ 6000 platform (Illumina). The PE reads were filtered using Cutadapt (v.1.9) and then quality-checked with FastQC (v.0.11.9). Mapped reads of each sample were assembled using StringTie (v.2.1.6) with default parameters, and transcriptomes were merged using gffcompare (v.0.9.8). Transcript abundance was estimated using Ballgown (http://www.bioconductor.org/packages/release/bioc/html/ballgown.html) and StringTie, calculating fragment per kilobase of transcript per million mapped reads (FPKM). Differential gene expression analysis was performed with DESeq2 (v.1.30.0) using default settings. Genes were considered DEG if they exhibited a |log2FC| > 0.58 and

*p*-value < 0.05. All experiments were conducted with three biological replicates.

## Statistical analysis

All experiments were repeated at least three time. The data are shown as mean ± SEM. Statistical analysis and graphical representation were conducted using GraphPad Prism. The statistical significance was determined using Student's t-test. Significance levels were denoted as follows: *$P < 0.05$, **$P < 0.01$, ***$P < 0.001$, ****$P < 0.0001$, and "ns" indicates non-significant ($P \geq 0.05$).

## Data availability

The data supporting the findings of this study are available from the corresponding author upon reasonable request. ChIP-seq, ATAC-seq, and RNA-seq data can be accessed in Dataset EV1, EV2, EV4. The datasets produced in this study are available in the following databases: [ChIP-seq, ATAC-seq, and RNA-seq]: [Gene Expression Omnibus database] [GSE291642 (private status: yxuvyckgjtchjmt)] (https://www.ebi.ac.uk/biostudies/studies/S-BSST1949?key=d1bc9743-a59b-443a-9cee-77be410f2508). The raw data can be found in the raw data file or database. [raw data of Fig. 7]: [BioStudies database] [S-BSST1949] (https://www.ebi.ac.uk/biostudies/studies/S-BSST1949?key=d1bc9743-a59b-443a-9cee-77be410f2508).

The source data of this paper are collected in the following database record: biostudies:S-SCDT-10_1038-S44321-025-00247-x.

## Peer review information

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

## Acknowledgements

We extend our heartfelt thanks to Professor Taosheng Huang for providing the WES data and to Professor Ying Feng for supplying the *Myod1*-Cre mouse model and offering invaluable scientific guidance. We also wish to express our sincere gratitude to all members of the pedigree for their active cooperation in advancing scientific research. This work was supported by the National Natural Science Foundation of China (82271889 to J Ma, 82371173 to T Zhang, 32170885 and 32370939 to B Hao), the National Key Research and Development Program of China (2021YFC2701001 and 2021YFC2700803 to D Ma), Shanghai Natural Science Foundation (23ZR1409400 to J Ma and 24ZR1409400 to XC), and Shanghai Technical Standards Project (21DZ2200700 to T Zhang). We thank BioRender for providing the tools to create scientific illustrations (under a paid subscription) (https://BioRender.com).

## Author contributions

**Peixuan Wu**: Data curation; Validation; Investigation; Visualization; Methodology; Writing—original draft. **Nan Song**: Resources; Project administration. **Yang Xiang**: Formal analysis; Validation; Investigation; Visualization; Methodology. **Zhe Tao**: Resources. **Bing Mao**: Resources. **Ruochen Guo**: Resources. **Xin Wang**: Resources. **Dan Wu**: Resources. **Zhenzhen Zhang**: Resources. **Xin Chen**: Resources; Funding acquisition. **Duan Ma**: Conceptualization; Supervision; Funding acquisition; Project administration. **Tianyu Zhang**: Conceptualization; Supervision; Funding acquisition; Project administration. **Bingtao Hao**: Conceptualization;

Supervision; Funding acquisition; Project administration; Writing—review and editing. **Jing Ma**: Conceptualization; Supervision; Funding acquisition; Project administration; Writing—review and editing.

Source data underlying figure panels in this paper may have individual authorship assigned. Where available, figure panel/source data authorship is listed in the following database record: biostudies:S-SCDT-10_1038-S44321-025-00247-x.

## Disclosure and competing interests statement

The authors declare no competing interests.

# Expanded View Figures

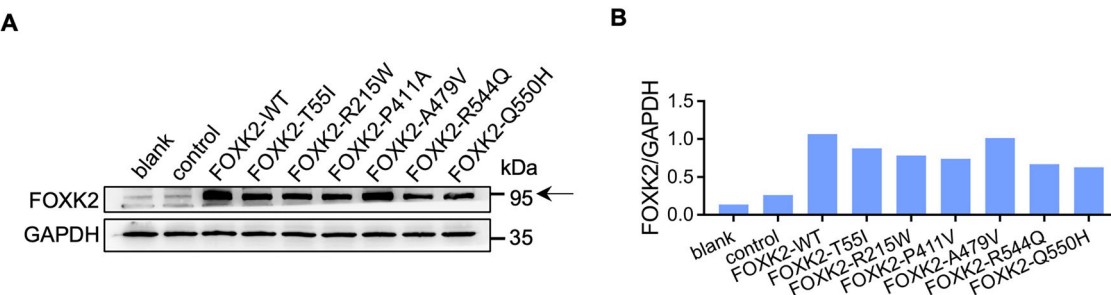

**Figure EV1. The protein expression of FOXK2 mutants in HEK-293T cells.**

(A, B) Western blot analysis of FOXK2 expression level in HEK-293T cells. The black arrow points to specific protein band. The statistical analysis of FOXK2 protein expression level relative to GAPDH is shown in (B).

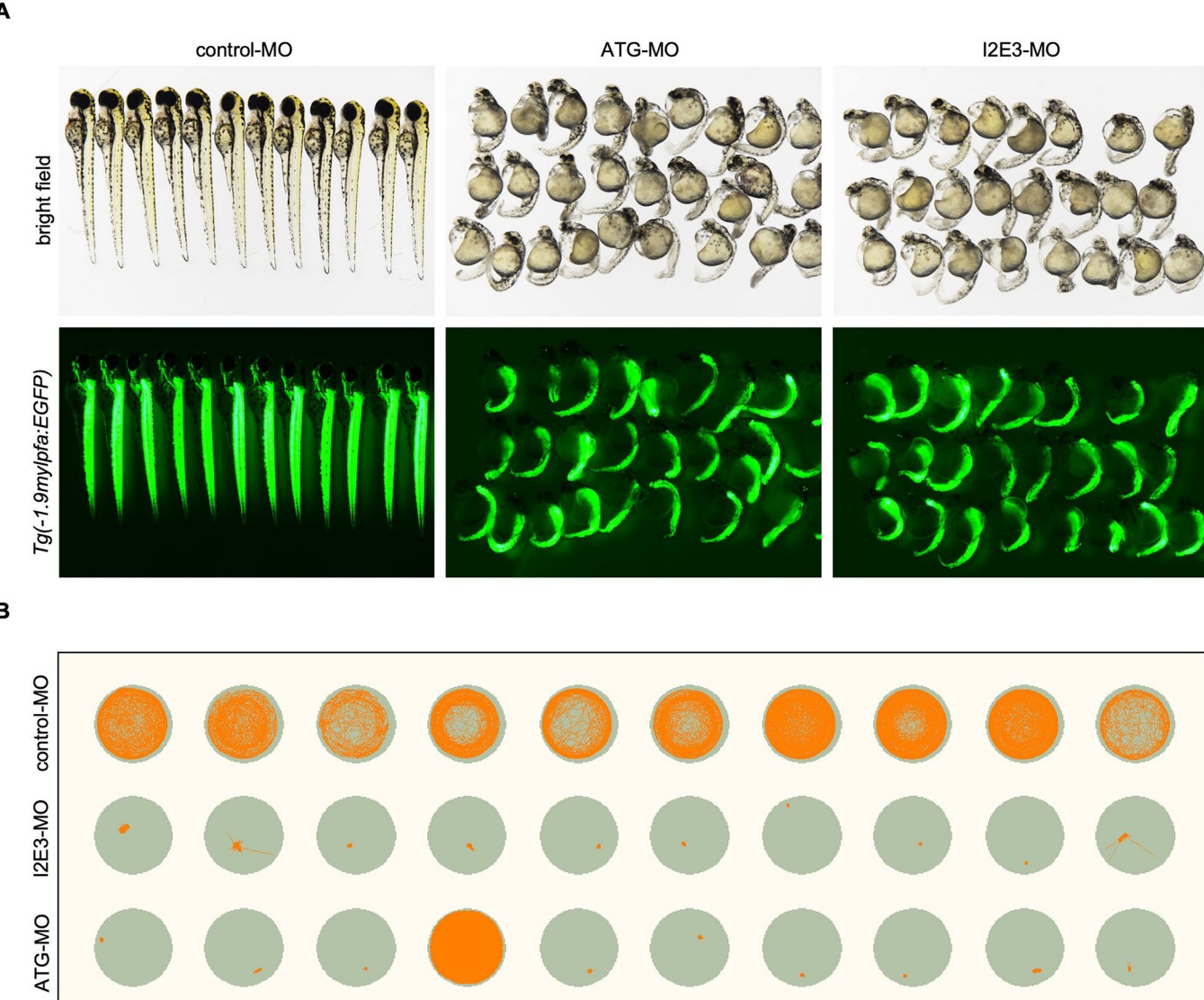

**Figure EV2. Gross morphology and locomotor capacity of *foxk2* morphants.**

(A) Gross morphology of *Tg(-1.9mylpfa:EGFP)* zebrafish larvae at 72 hpf. (B) Digital movement trajectories of zebrafish larvae at 5 dpf. One embryo was placed per well (*n* = 10 per group). MO morpholino, hpf hours post fertilization, dpf days post fertilization.

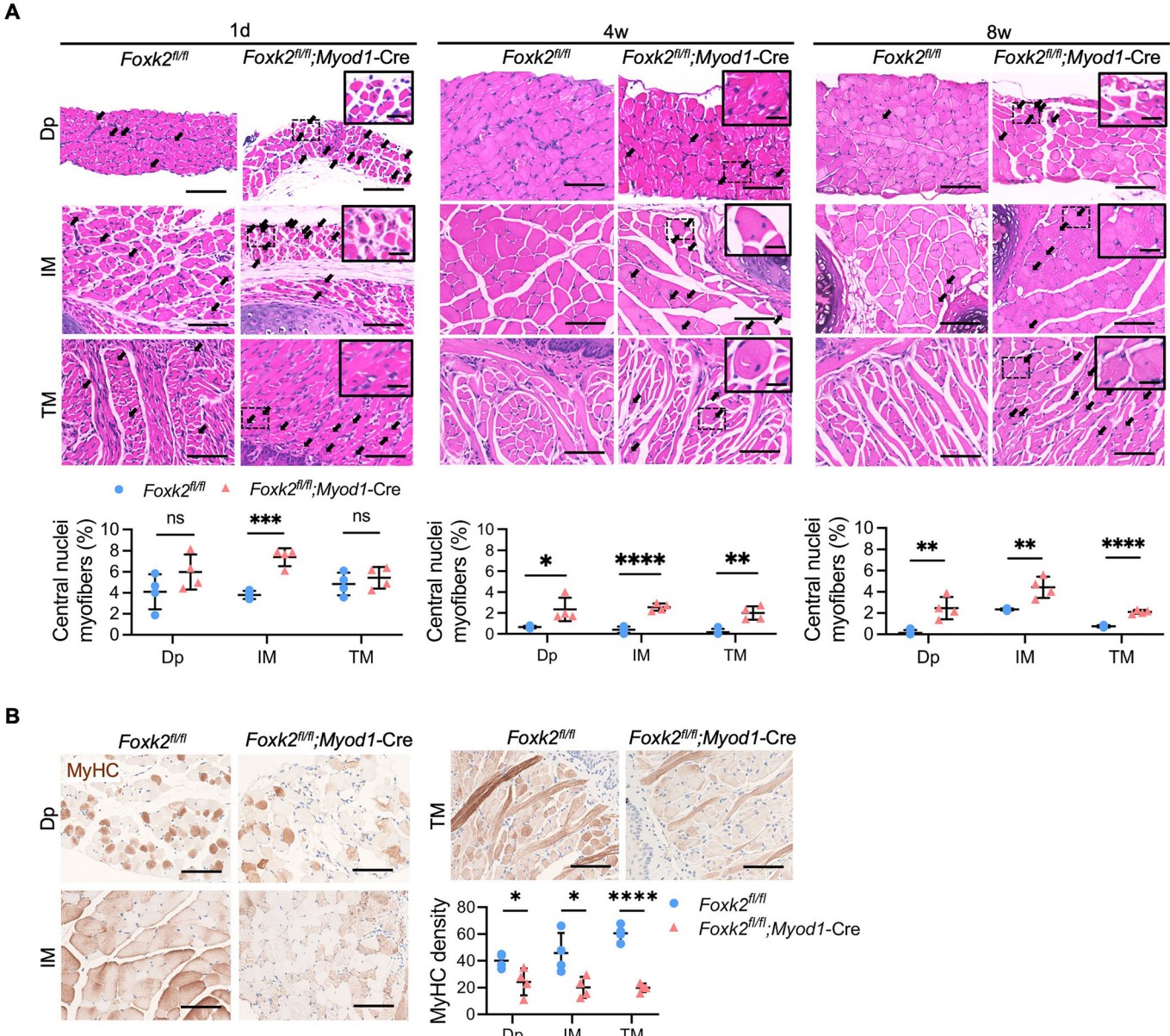

**Figure EV3. Abnormal skeletal muscle development in muscle stem cells-specific *Foxk2* conditional knockout mice.**

(A) H&E staining illustrating histological aspects and the percentage of central nuclei myofibers in the skeletal muscles (Dp, IM, and TM) of *Foxk2*fl/fl and *Foxk2*fl/fl; *Myod1*-Cre littermates at postnatal day 1, 4 weeks, and 8 weeks (*n* = 4 areas). Black arrows indicate central nuclei. The inset in the upper right corner (scale bars: 25 μm) is an enlarged view of the boxed area in the main image, highlighting central nuclei myofibers (scale bars: 100 μm). *P* value: 1 d: Dp, ns-*P* = 0.1619; IM, ****P* = 0.0003; TM, ns-*P* = 0.4606. 4w: Dp, **P* = 0.0244; IM, *****P* < 0.0001; TM, ***P* = 0.0023. 8w: Dp, ***P* = 0.0051; IM, ***P* = 0.0057; TM, *****P* < 0.0001. (B) IHC staining of MyHC in the skeletal muscles (Dp, IM, TM) of *Foxk2*fl/fl and *Foxk2*fl/fl; *Myod1*-Cre littermates at 8 weeks (*n* = 4 areas). MyHC was quantified by InDen/Area using ImageJ software. scale bars: 100 μm. *P* value: Dp, **P* = 0.0332; IM, **P* = 0.0238; TM, *****P* < 0.0001. Data were analyzed by Student's t test. All error bars indicate mean ± standard deviation. d day, w week, Dp diaphragm, IM intercostal muscle, TM tongue muscle, H&E hematoxylin-eosin staining, IHC immunohistochemistry.

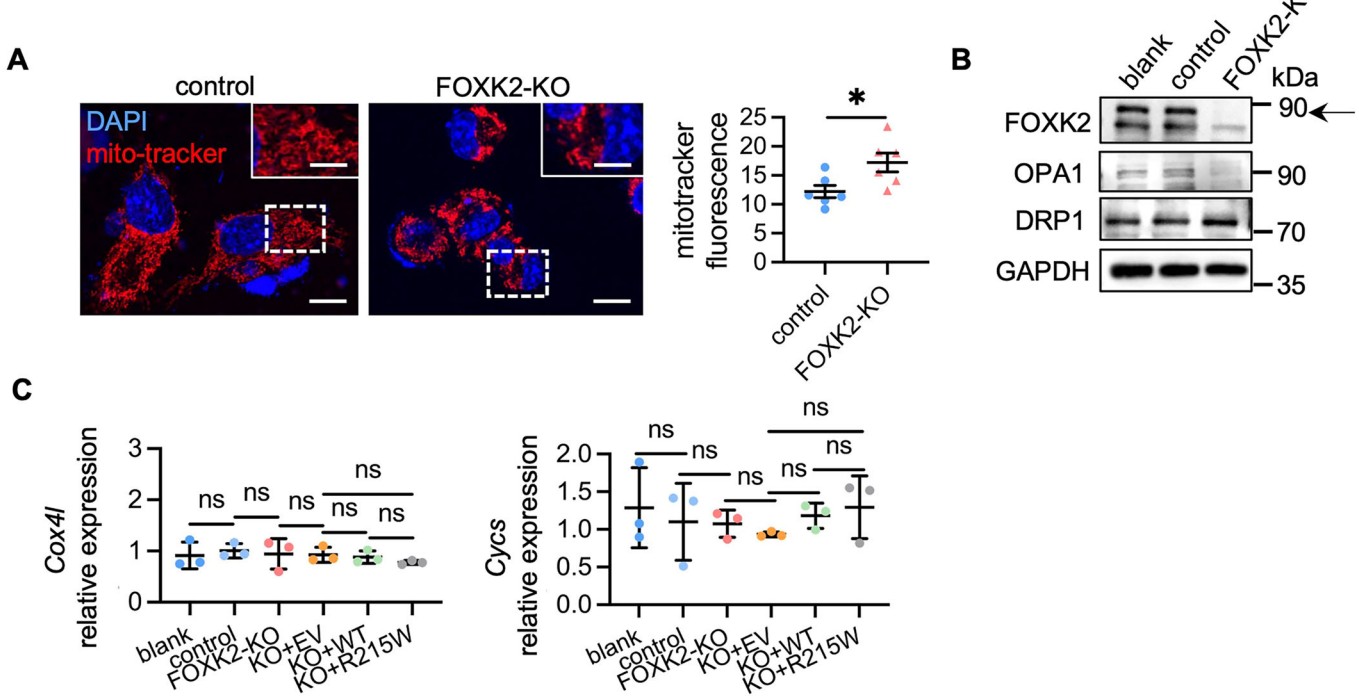

**Figure EV4. Mitochondrial morphology and mitochondrial function-related gene expression in C2C12 cells.**

(A) Confocal images of mitochondria labeled with mitotracker (red) and nuclei stained with DAPI (blue). The inset in the upper right corner provides an enlarged view (scale bars: 8 μm) of the boxed area in the main image, highlighting mitochondria (scale bars: 20 μm). Fluorescence intensity was quantified using ImageJ software ($n = 6$ areas). P value: *$P = 0.0281$. (B) Western blot analysis showing the expression levels of OPA1 and DRP1 in C2C12 cells. The black arrow indicates the specific protein band. (C) Quantitative of relative mitochondrial gene expression levels in C2C12 cells by qPCR ($n = 3$ repeats). Data were analyzed by Student's t test. All error bars indicate mean ± standard deviation. Blank untreated wild-type C2C12 cells, control C2C12 cells transfected with a blank vector as a control for the FOXK2-KO group, FOXK2-KO FOXK2 knockout C2C12 cells, KO + EV FOXK2-KO cells transfected with a blank vector as a control for KO + WT and KO + R215W groups, KO + WT FOXK2-KO cells with overexpression of the human wild-type FOXK2 vector, KO + R215W FOXK2-KO cells with overexpression of the human FOXK2 R215W mutation vector, qPCR quantitative real-time polymerase chain reaction.

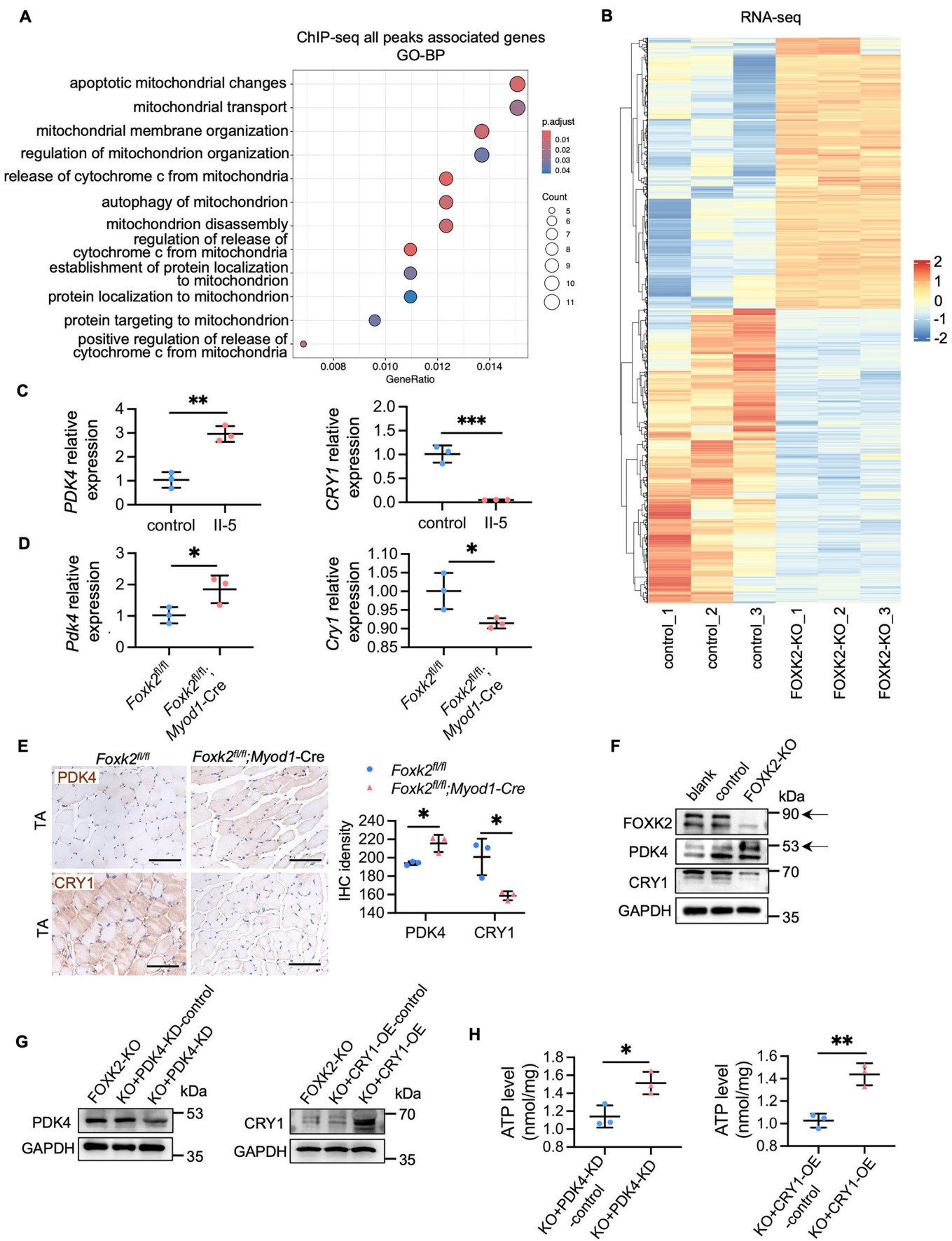

**Figure EV5.   Chromatin immunoprecipitation sequencing and RNA sequencing analysis with target gene validation in FOXK2-KO C2C12 cells.**

(A) GO analysis (BP) of all genes of which peaks bound by FOXK2, focusing on mitochondrion-related terms. (B) Heatmap of DEGs based on RNA-seq in FOXK2-KO C2C12 cells, with upregulated genes shown in red and downregulated genes in blue. (C) Quantitative of *PDK4* and *CRY1* gene expression levels in the EM of the proband and a control from pedigree1 by qPCR ($n = 3$ repeats). P value: ***$P = 0.0008$; **$P = 0.0020$. (D) Quantitative of *Pdk4* and *Cry1* gene expression levels in TA muscles of *Foxk2*<sup>fl/fl</sup> and *Foxk2*<sup>fl/fl</sup>; *Myod1*-Cre littermates at 8 weeks by qPCR ($n = 3$ repeats). P value: *Pdk4*, *$P = 0.0487$; *Cry1*, *$P = 0.0418$. (E) IHC staining of PDK4 and CRY1 in the TA muscles of *Foxk2*<sup>fl/fl</sup> and *Foxk2*<sup>fl/fl</sup>; *Myod1*-Cre littermates at 8 weeks ($n = 3$ areas). The intensity of PDK4 and CRY1 was quantified by InDen/Area using ImageJ software. Scale bars: 100 μm. P value: PDK4, *$P = 0.0174$; CRY1, *$P = 0.0233$. (F) Western blot analysis of PDK4 and CRY1 expression level in C2C12 cells. The black arrows indicate the specific protein bands. (G) Western blot analysis of PDK4 knockdown and CRY1 overexpression levels in FOXK2-KO C2C12 cells. (H) Quantification of ATP contents in C2C12 cells ($n = 3$ repeats). Values were normalized to the total cellular protein level. P value: **$P = 0.0064$; *$P = 0.024$. Data were analyzed by Student's t test. All error bars indicate mean ± standard deviation. ChIP-seq chromatin immunoprecipitation sequencing, RNA-seq RNA sequencing, GO Gene Ontology, BP biological process, DEGs differentially expressed genes, EM eyelid muscle, TA tibialis anterior, qPCR quantitative real-time polymerase chain reaction, IHC immunohistochemistry, blank untreated wild-type C2C12 cells, control C2C12 cells transfected with a blank vector as a control for the FOXK2-KO group, FOXK2-KO FOXK2 knockout C2C12 cells, KO + PDK4-KD-control FOXK2-KO C2C12 cells transfected with a blank vector as a control for the KO + PDK4-KD group, KO + PDK4-KD FOXK2-KO C2C12 cells transfected with a PDK4 knockdown plasmid, KO + CRY1-OE-control FOXK2-KO C2C12 cells transfected with a blank vector as a control for the KO + CRY1-OE group, KO + CRY1-OE FOXK2-KO C2C12 cells transfected with a CRY1 overexpression plasmid.

