## [Peer Review File · EMBO Molecular Medicine]

FO XK2 in Skeletal Muscle Development: A New Pathogenic Gene for Congenital Myopathy with Ptosis

Jing Ma, Peixuan Wu, Nan Song, Yang Xiang, Zhe Tao, Bing Mao, Ruochen Guo, Xin Wang, Dan Wu, Zhenzhen Zhang, Xin Chen, Duan Ma, Tianyu Zhang, and Bingtao Hao

Corresponding authors: Jing Ma (maj14@fudan.edu.cn), Duan Ma (duanma@fudan.edu.cn), Tianyu Zhang (ty_zhang@fudan.edu.cn), Bingtao Hao (haobt123@zzu.edu.cn)

Review Timeline:

Submission Date:	21st Dec 24
Editorial Decision:	10th Jan 25
Revision Received:	6th Apr 25
Editorial Decision:	15th Apr 25
Revision Received:	26th Apr 25
Accepted:	29th Apr 25

Editor: *Zeljko Durdevic*

Transaction Report:

10th Jan 2025

Dear Dr. Ma,

Thank you for the submission of your manuscript to EMBO Molecular Medicine. We have now received feedback from the three reviewers who agreed to evaluate your manuscript. All three referees recognize interest of the study but also raise important concerns that should be addressed in a major revision. If you would like to discuss further the points raised by the referees, I am available to do so via email or video. Let me know if you are interested in this option.

Further consideration of a revision that addresses reviewers' concerns in full will entail a second round of review. EMBO Molecular Medicine encourages a single round of revision only and therefore, acceptance or rejection of the manuscript will depend on the completeness of your responses included in the next, final version of the manuscript. For this reason, and to save you from any frustrations in the end, I would strongly advise against returning an incomplete revision. Further, when submitting the revised manuscript please be sure to add institutional email addresses for Jing Ma, Tianyu Zhang and Bingtao Hao in our submission system.

We would welcome the submission of a revised version within three months for further consideration. Please let us know if you require longer to complete the revision.

I look forward to receiving your revised manuscript.

Yours sincerely,

Zeljko Durdevic

We require:

- 1) A .docx formatted version of the manuscript text (including legends for main figures, EV figures and tables). Please make sure that the changes are highlighted to be clearly visible.
- 2) Individual production quality figure files as .eps, .tif, .jpg (one file per figure). For guidance, download the 'Figure Guide PDF': (<https://www.embopress.org/page/journal/17574684/authorguide#figureformat>).
- 3) A .docx formatted letter INCLUDING the reviewers' reports and your detailed point-by-point responses to their comments. As part of the EMBO Press transparent editorial process, the point-by-point response is part of the Review Process File (RPF), which will be published alongside your paper.
- 4) A complete author checklist, which you can download from our author guidelines (<https://www.embopress.org/page/journal/17574684/authorguide#submissionofrevisions>). Please insert information in the checklist that is also reflected in the manuscript. The completed author checklist will also be part of the RPF.
- 5) Please note that all corresponding authors are required to supply an ORCID ID for their name upon submission of a revised manuscript.

6) It is mandatory to include a 'Data Availability' section after the Materials and Methods. Before submitting your revision, primary datasets produced in this study need to be deposited in an appropriate public database, and the accession numbers and database listed under 'Data Availability'. Please remember to provide a reviewer password if the datasets are not yet public (see <https://www.embopress.org/page/journal/17574684/authorguide#dataavailability>).

12) Author contributions: You will be asked to provide CRediT (Contributor Role Taxonomy) terms in the submission system. These replace a narrative author contribution section in the manuscript.

13) A Conflict of Interest statement should be provided in the main text.

14) Every published paper now includes a 'Synopsis' to further enhance discoverability. Synopses are displayed on the journal webpage and are freely accessible to all readers. They include a short stand first (maximum of 300 characters, including space) as well as 2-5 one-sentences bullet points that summarizes the paper. Please write the bullet points to summarize the key NEW

findings. They should be designed to be complementary to the abstract - i.e. not repeat the same text. We encourage inclusion of key acronyms and quantitative information (maximum of 30 words / bullet point). Please use the passive voice. Please attach these in a separate file or send them by email, we will incorporate them accordingly.

15) Include a Reagents and Tools Table as part of the Methods section, which can be downloaded from our author guidelines (<https://www.embopress.org/page/journal/17574684/authorguide#structuredmethods>)

**** Reviewer's comments ****

Referee #1 (Comments on Novelty/Model System for Author):

The applicant uses multiple models that are appropriate: human patient samples, zebrafish morphants/transgenic lines, and conditional knockout mouse models.

Referee #1 (Remarks for Author):

The manuscript by Wu et al., centers on the characterization of FOXK2 as a novel regulator of skeletal muscle development and a causative gene for a congenital myopathy. The authors identified mutations in six pedigrees with an undefined congenital myopathy and ptosis through whole exome sequencing and Sanger sequencing. Zebrafish foxk2 morphant analysis revealed muscle damage in early development using a variety of transgenic models and locomotor analysis. Foxk2 MyoD1 Cre conditional KO mice revealed muscle structural defects and mitochondria respiration defects. ChIP-seq and ATAC-seq analysis identified OXPHOS and other mitochondria-related FOXK2 target genes. The authors concluded that FOXK2 is a novel pathogenic gene for congenital myopathy with ptosis and highlights its essential role in skeletal muscle development and mitochondrial homeostasis

Overall, the manuscript is well-written and comprehensive in the analysis of Foxk2's function in skeletal muscle and their affected FOXK2 patients. I have a few minor/moderate comments to clarify the exact role of FOXK2 in muscle development. The statistical analysis is appropriate and the sample size is also appropriate for the experiments. This manuscript could be a valuable addition to elucidating the role of forkhead transcription factors/FOXK2 in muscle development and human disease.

General Comments:

1. A bit more background on the role of other forkhead transcription factors in muscle development and disease is warranted. Foxk1 (formerly myocyte nuclear factor/MNF) has been implicated as a regulator of muscle progenitor cells and regeneration via cyclins (Garry et al., PNAS 2000; Hawke et al., JBC, 2003).
2. It appears that the Foxk2 MyoD1 KO mice have impaired activated muscle progenitor cells (MPC) due to mitochondria defects and activated ROS. Have the authors evaluated MPC cell death in this model (e.g. TUNEL or Annexin staining)? Does the increased ROS lead towards a reduced number of MPCs in this Foxk2 MyoD1 KO model?
3. Did the myotube size increase in the Foxk2 KO experiments conducted in Figure 5?
4. The Sukonina et al., Nature 2019 study implicated both FOXK1 and FOXK2 in glycolysis. Did the authors evaluate glucose and/or glycolysis function in their patients and/or mouse model?
5. Does mitochondria number change in the Foxk2 MyoD1 KO mouse muscles? My question centers more on the mechanism of correction via Co-enzyme Q10 (Figure 7 experiments).
6. Minor comment. Just to clarify, there was no phenotype observed in the Foxk2+/flox MyoD1 heterozygote mice? E.g. there is no autosomal dominant or hypomorphic phenotypes/effects observed in the mice similar to that of the FOXK2 patients.

Referee #2 (Remarks for Author):

This work by Ma et al. has significantly contributed to our understanding of the etiology of myopathy in congenital ptosis,

particularly in elucidating the novel function of FOXC2 in skeletal muscle development. Currently, there are only a handful of studies on isolated congenital ptosis, lacking solid data to confirm the underlying pathogenic factors. The findings of the current research are definitely invaluable as multiple new pathogenic genetic factors were identified for isolated congenital ptosis, with support from clinical data to animal models, cell models, and molecular mechanism studies. The data seemed solid and credible. Interestingly, the author proposes that FOXC2 is also a causative factor in congenital myopathy associated with ptosis, providing new insights for research in this area. Overall, this work is well-designed work, with high-quality data and proper interpretation. I would recommend publication of this work, provided that the following points are fully addressed.

Major points:

1. Patients of clinical family collected by the authors had ptosis. However, it is not stated whether the FOXC2 knockout mice have ptosis phenotype, so the authors should explain it.
2. Based on the data from multi-omics analysis, the authors found the direct target genes PDK4 and CRY1 and verified their mRNA levels in both C2C12 and MuSC cells. What about their proteins? It is necessary to examine the changes in both mRNA and protein levels of PDK4 and CRY1 in mouse models and the clinical patients as well.
3. Furthermore, the authors need to provide further data on the possible phenotypes when rescuing PDK4 and CRY1 in FOXC2-KO cells to validate your viewpoints.
4. Are there any correlations between the discovered mutation in the FOXC2 gene and the decrease of FOXC2 expression in the eyelid muscles of the proband in the first family? Did the authors construct a mouse model of the FOXC2 gene mutation?

Minor:

1. The authors observed short stature and reduced exercise capacity in FOXC2 knockout mice. Is this apparent phenotype stemming from a skeletal developmental defect? Can the authors comment on it?
2. The authors identified the effect of FOXC2 deficiency on the expression of mitochondrial function-related genes in muscle stem cells and C2C12 cells. Why different sets of genes were chosen? This reviewer would like to see the data currently missing for the other genes.
3. The authors singled out the Pdk4 and Cry1 genes based on ATAC-seq without showing their respective peak plots. It would be better to provide these data.
4. This study provided convincing data demonstrating that FOXC2 is essential for skeletal muscle development. Meanwhile, the differential expression patterns were observed for FOXC1 and FOXC2 in myogenesis of embryonic mice and myogenic differentiation of C2C12 cells (Figure S6), suggesting that FOXC1 and FOXC2 might be of distinct functions. Is there any concern about the function of FOXC1 in skeletal muscle? The authors need to discuss in appropriate places .
5. The annotation of some figures in the article is not consistent in styles. For example, the annotation of figure 1b "H&E" is outside the diagram and figure 6b "DAPI DRP1" is in the diagram, so it is recommended that the author modify it.
6. Lastly, language should be further polished , which would tremendously benefit the readership. For instance: "pave the way for improved strategies..."

Referee #3 (Remarks for Author):

With great interest, I have read the manuscript by Wu et al, which identifies FOXC2 as a novel pathogenic gene associated with congenital myopathy and ptosis. Through whole exome sequencing (WES) of clinical families and functional studies in animal and cell models, the research reveals the role of FOXC2 in regulating skeletal muscle development. FOXC2 is shown to maintain mitochondrial function, which is crucial for the development of muscle stem cells (MuSCs). The study also provides a new perspective on the genetic etiology of congenital ptosis and has broader implications for disorders related to skeletal muscle development.

This is a well written and performed study, that should be published once my comments are suitable addressed:

1. It remains unclear why the authors consider the three variants (P411A, A479V, R544Q) from pedigrees 3, 4, and 5 to be pathogenic rather than rare polymorphisms. Firstly, for a congenital, dominant rare disease, the frequency of these variants in the control population seems somewhat high. Secondly, upon screening these three variants (P411A, A479V, R544Q) in our local whole-exome sequencing (WES) database, which includes over 2000 unique patients, we found that these variants were present in several individuals without myopathy and ptosis. Thirdly, while in silico predictions suggest that all these variants are likely damaging, further experiment need to be identified to ascertain pathogenicity, such as their impact on protein expression.
2. Why does the heterozygous variant p.Arg215Trp result in a reduction of protein levels to an undetectable level (figure 1D)? Additional evidence is required to ascertain the loss of function (LoF), such as the use of point mutation cell models.
3. The pedigree families 3, 4, and 5 exhibit dominant traits. Are there additional members within these families available for genetic segregation analysis?
4. The detailed phenotypes for each pedigree should be provided, including age at onset, muscle MRI, and muscle histopathological examination. The index patient in pedigrees 3-5 exhibits global developmental delay. What are the phenotypes of their parents? I believe they appear to be milder carriers, as they are typically married and childbearing; Why do phenotypes vary within the same family and across different families (isolated ptosis in pedigree 1)?
5. Western blot shows that FOXC2 has two bands in Figure 5A,C but only one band in figure 1D, which one is the true target

band? The expression level of the R275W mutant plasmid((Figure5C)) is not lower than that of WT, and may even be higher, which is inconsistent with the results of LoF in figure 1D.

6.It is recommended to provide quantification of mitochondrial morphology through confocal imaging of labeled mitochondria. Additionally, Western blot analysis of key proteins involved in mitochondrial dynamics should also be supplied.

Responses to the reviewers' comments:**Referee #1**

The manuscript by Wu et al., centers on the characterization of FO XK2 as a novel regulator of skeletal muscle development and a causative gene for a congenital myopathy. The authors identified mutations in six pedigrees with an undefined congenital myopathy and ptosis through whole exome sequencing and Sanger sequencing. Zebrafish *foxk2* morphant analysis revealed muscle damage in early development using a variety of transgenic models and locomotor analysis. *Foxk2* *MyoD1* Cre conditional KO mice revealed muscle structural defects and mitochondria respiration defects. ChIP-seq and ATAC-seq analysis identified OXPHOS and other mitochondria-related FO XK2 target genes. The authors concluded that FO XK2 is a novel pathogenic gene for congenital myopathy with ptosis and highlights its essential role in skeletal muscle development and mitochondrial homeostasis.

Overall, the manuscript is well-written and comprehensive in the analysis of FO XK2's function in skeletal muscle and their affected FO XK2 patients. I have a few minor/moderate comments to clarify the exact role of FO XK2 in muscle development. The statistical analysis is appropriate and the sample size is also appropriate for the experiments. This manuscript could be a valuable addition to elucidating the role of forkhead transcription factors/FO XK2 in muscle development and human disease.

General Comments:

1. A bit more background on the role of other forkhead transcription factors in muscle development and disease is warranted. FO XK1 (formerly myocyte nuclear factor/MNF) has been implicated as a regulator of muscle progenitor cells and regeneration via cyclins (Garry et al., PNAS 2000; Hawke et al., JBC, 2003).

Response:

Thank you for your valuable suggestion. We have already discussed the role of other FOX family members in skeletal muscle development in lines 646-661 of the discussion section. In response to your comment, we have now included additional information on FO XK1 (formerly myocyte nuclear factor, MNF), based on the references you

provided. Specifically, we have added the following statement in lines 659-661:

"FOXK1, a homolog of FOXK2, regulates the myogenic progenitor cell population essential for skeletal muscle regeneration. FOXK1 plays a critical role in the proliferation and differentiation of MuSCs following muscle injury and serves as a molecular marker for the quiescent myogenic progenitor cell population."

We believe these additions help deepen the reader's understanding of the Forkhead box transcription factor family's involvement in skeletal muscle development. We hope these revisions address your concerns. Thank you again for your insightful feedback.

2. It appears that the *Foxk2 MyoD1* KO mice have impaired activated muscle progenitor cells (MPC) due to mitochondria defects and activated ROS. Have the authors evaluated MPC cell death in this model (e.g. TUNEL or Annexin staining)? Does the increased ROS lead towards a reduced number of MPCs in this *Foxk2 MyoD1* KO model?

Response:

Thank you for your insightful comment. It is well established that prolonged exposure to reactive oxygen species (ROS) can induce apoptosis in stem cell populations, including embryonic stem cells^(Guo et al, 2010). To assess whether elevated ROS levels in skeletal muscle tissue of the *Foxk2^{fl/fl}; Myod1-Cre* mice impact muscle stem cells (MuSCs) death, we performed an immunofluorescence assay to detect apoptosis in MuSCs. In these assays, MuSCs were labeled with PAX7, and apoptotic cells were identified using TUNEL staining (scale bar = 100 μ m). Our data show a significant increase in TUNEL-positive MuSCs in the *Foxk2^{fl/fl}; Myod1-Cre* mice compared to the control group. We hope this additional data addresses your concerns regarding the effect of ROS on MuSCs survival in this model. Thank you again for your helpful feedback, which has contributed to the improvement of our manuscript.

3. Did the myotube size increase in the FOXK2 KO experiments conducted in Figure 5?

Response:

Thank you for your careful attention to detail. To more accurately reflect the myogenic differentiation capacity of each group, we have added a statistical analysis of myofiber diameter (measured in μm , with 23 myofibers per group) in Figure 5E. and we have updated the results section to state:

"In FOXK2-KO C2C12 cells, the typical long and well-organized multinucleated myotubes observed in control cells were replaced by fewer, shorter, thinner, and disorganized myotubes. Importantly, introducing wild-type FOXK2 protein restored normal myotube morphology, whereas the mutated FOXK2 protein did not."

These updates are now reflected in line 545-547. We hope this additional information provides a clearer picture of the myotube size and morphology changes observed in the FOXK2-KO experiments. Thank you again for your valuable feedback, which has significantly improved the clarity and accuracy of our manuscript.

4. The Sukonina et al., Nature 2019 study implicated both FOXK1 and FOXK2 in glycolysis. Did the authors evaluate glucose and/or glycolysis function in their patients and/or mouse model?

Response:

Thanks for your helpful suggestion and reminder. As noted, FOXK2 has been identified as a positive regulator of glycolysis in multiple myeloma cells(Liu *et al*, 2023) and adipocytes(Sukonina *et al*, 2019). In response to your comment, we investigated the role of FOXK2 in glycolysis regulation in skeletal muscle by measuring glucose content in gastrocnemius (Gas) and tibialis anterior muscle (TA) muscle tissues of *Foxk2^{fl/fl}* and *Foxk2^{fl/fl}; Myod1-Cre* mice. We found that glucose accumulation in the Gas and TA muscles of the *Foxk2^{fl/fl}; Myod1-Cre* mice was higher compared to the control group, suggesting that glucose metabolism was impaired due to the knockout of *Foxk2*. This supports the positive role of FOXK2 in regulating glycolysis in skeletal muscle.

However, since our study does not focus on this aspect, we chose not to include these data in this article. We agree that further experimental investigation is needed, and your suggestion provides an important direction for future research. We hope this addition addresses your concern and clarifies our study. Thank you once again for your valuable input.

5. Does mitochondria number change in the *Foxk2 MyoD1* KO mouse muscles? My question centers more on the mechanism of correction via Co-enzyme Q10 (Figure 7 experiments).

Response:

Thank you for your insightful suggestion. Our results show that, prior to treatment, *Foxk2^{fl/fl}; Myod1-Cre* mice exhibit an higher number of damaged mitochondria compared to *Foxk2^{fl/fl}* controls. Following Coenzyme Q10 (CoQ10) treatment, we observed a reduction in the number of damaged mitochondria, along with a restoration of mitochondrial morphology, characterized by well-defined membrane structures and intact cristae. These results are now reflected in Figure 7D, and the following explanation has been added to line 576-578:

"CoQ10 treatment improved mitochondrial morphology, resulting in a greater

number of mitochondria with well-defined membrane structures and intact mitochondrial cristae."

We hope these revisions help clarify the mechanisms by which CoQ10 treatment corrects mitochondrial dysfunction. Thank you once again for your careful review and valuable feedback.

6. Minor comment. Just to clarify, there was no phenotype observed in the *Foxk2^{+fllox}* MyoD1 heterozygote mice? E.g. there is no autosomal dominant or hypomorphic phenotypes/effects observed in the mice similar to that of the FOXK2 patients.

Response:

Thank you for your careful attention to detail. We have already mentioned the issue in lines 461-462:

"Heterozygous mice (*Foxk2^{+fl}*; *Myod1-Cre*) did not show any noticeable abnormal phenotypes."

For example, when examining the proportion of central nuclei myofibers, we found no significant difference in representative muscles (gastrocnemius, tibialis anterior, diaphragm, intercostal muscles, tongue muscle) between *Foxk2^{+fl}*; *Myod1-Cre* and *Foxk2^{fl/fl}* mice. The absence of a distinct phenotype in the heterozygous knockout mice is likely due to the challenges of fully mimicking the human phenotype in mice, given the differences in genetic backgrounds between humans and mice.

We appreciate your thoughtful comments and professional advice, which have significantly improved the accuracy and rigor of our manuscript. Thank you again for your valuable contribution.

Referee #2

This work by Ma et al. has significantly contributed to our understanding of the etiology of myopathy in congenital ptosis, particularly in elucidating the novel function of FO XK2 in skeletal muscle development. Currently, there are only a handful of studies on isolated congenital ptosis, lacking solid data to confirm the underlying pathogenic factors. The findings of the current research are definitely invaluable as multiple new pathogenic genetic factors were identified for isolated congenital ptosis, with support from clinical data to animal models, cell models, and molecular mechanism studies. The data seemed solid and credible. Interestingly, the author proposes that FO XK2 is also a causative factor in congenital myopathy associated with ptosis, providing new insights for research in this area. Overall, this work is well-designed work, with high-quality data and proper interpretation. I would recommend publication of this work, provided that the following points are fully addressed.

Major points:

1. Patients of clinical family collected by the authors had ptosis. However, it is not stated whether the FO XK2 knockout mice have ptosis phenotype, so the authors should explain it.

Response:

Thank you for your thoughtful comment and for taking the time to review our manuscript. As you pointed out, the clinical family studied by the authors had patients with ptosis. However, it is difficult to observe ptosis in *Foxk2* knockout mice due to the differences in the frequency of blinking between humans and rodents. Unlike humans, who blink every few seconds, rodents blink much less frequently (Turner *et al*, 2023), which makes it challenging to assess the presence of ptosis in mice. Instead, we evaluated the development of the eyelid muscles in *Foxk2^{fl/fl}; Myod1-Cre* mice by examining histological sections. Compared to the *Foxk2^{fl/fl}* control group, we observed a significant increase in the proportion of central nuclei myofibers in the eyelid muscles of the *Foxk2^{fl/fl}; Myod1-Cre* mice (Figure 3E). This finding suggests impaired skeletal muscle development, although it does not directly translate into a ptosis phenotype. We

discussed this observation in the manuscript as follows:

"This lack of ptosis may result from the infrequent use of eyelids in mice, which could minimize the pathological impact on their movement" (lines 693-695).

We hope this explanation addresses your concern regarding the absence of ptosis in the *Foxk2* knockout mice. Thank you again for your careful review and valuable feedback.

2. Based on the data from multi-omics analysis, the authors found the direct target genes PDK4 and CRY1 and verified their mRNA levels in both C2C12 and MuSC cells. What about their proteins? It is necessary to examine the changes in both mRNA and protein levels of PDK4 and CRY1 in mouse models and the clinical patients as well.

Response:

Thank you for your valuable suggestion. We agree that examining the protein levels of PDK4 and CRY1 in addition to their mRNA levels strengthens our findings. In C2C12 cells, we performed western blot analysis and found that PDK4 protein expression was upregulated, while CRY1 protein expression was downregulated following FOXK2 knockout compared to the control group (specific protein bands are indicated with black arrows). These results are now included in Figure EV5F.

For the clinical samples, we used RT-qPCR to measure the mRNA levels of *PDK4* and *CRY1* in the eyelid muscle tissues of the proband (II-5) from pedigree 1 (3 repeats). The results showed that *PDK4* expression was upregulated and *CRY1* expression was downregulated in the proband, which are now shown in Figure EV5C.

Additionally, we analyzed PDK4 and CRY1 expression in the tibialis anterior (TA) muscle of mice. In *Foxk2^{fl/fl}*; *Myod1-Cre* mice, the expression of PDK4 was significantly higher and the expression of CRY1 was significantly lower compared to *Foxk2^{fl/fl}* mice. These results are presented in Figure EV5D-E.

We have also added the following explanation to lines 624-626:

"We detected upregulation of PDK4 and downregulation of CRY1 in the EM tissue of the proband from pedigree 1. These findings were further validated in FOXK2-KO C2C12 cells, and TA tissues and MuSCs isolated from Foxk2^{fl/fl}; Myod1-Cre mice."

We hope these additions address your concern and strengthen our data. Thank you again for your careful review. We believe these updates improve the robustness of our findings.

3. Furthermore, the authors need to provide further data on the possible phenotypes when rescuing PDK4 and CRY1 in FOXK2-KO cells to validate your viewpoints.

Response:

We sincerely appreciate your insightful suggestion regarding the need for further validation of our findings through rescue experiments involving PDK4 and CRY1 in FOXK2-KO cells. We agree that such experiments will provide valuable insights into the functional consequences of FOXK2 deficiency and the potential roles of PDK4 and

CRY1 in mediating these effects. In response, we have designed rescue experiments to assess mitochondrial function by knocking down PDK4 and overexpressing CRY1 in FOXK2-KO cells. The statistical data show that the complementation of these two genes can restore ATP production. This result further confirms the correlation between the PDK4 /CRY1 genes, which are directly regulated by FOXK2, and the maintenance of mitochondrial function. These results are now presented in Figure EV5G-H, and the following explanation has been added to lines 627-629:

"Knockdown of PDK4 and overexpression of CRY1 in FOXK2-KO cells restored ATP production, further confirming the correlation between PDK4 / CRY1, genes directly regulated by FOXK2, and the maintenance of mitochondrial function."

We hope this addition addresses your concern and strengthens the validity of our findings. Thank you again for your valuable input.

4. Are there any correlations between the discovered mutation in the FOXK2 gene and the decrease of FOXK2 expression in the eyelid muscles of the proband in the first family? Did the authors construct a mouse model of the FOXK2 gene mutation?

Response:

Thank you for your insightful comment. To investigate whether the c.643C>T: p.R215W mutation affects FOXK2 expression, we overexpressed both wild-type FOXK2 (FOXK2-WT) and the mutant FOXK2 (FOXK2-R215W) plasmids in HEK-293T cells and assessed protein expression levels by western blot. The results showed that FOXK2 protein levels were reduced in cells expressing the R215W mutant compared to those expressing the wild-type protein, suggesting that the mutation leads to protein instability. These data are now included in Figure EV1, with the explanation:

" We further assessed these mutations through *in vitro* western blot experiments. The T55I, R215W, P411A, R544Q and Q550H mutations resulted in reduced FOXX2 protein expression." (lines 407–409).

We agree that a mouse model carrying the point mutation would provide valuable insight into the *in vivo* effects of this mutation. We are currently in the process of constructing such a model using CRISPR/Cas9-mediated gene editing, followed by breeding and phenotypic validation. However, due to the extensive time and resources required, the results from this model cannot be included in the current manuscript. That said, we believe our current findings lay a solid foundation for future in-depth studies using the point mutation model, which will help to further elucidate the role of FOXX2 in skeletal muscle development and disease. Thank you again for your thoughtful feedback.

Minor:

1. The authors observed short stature and reduced exercise capacity in FOXX2 knockout mice. Is this apparent phenotype stemming from a skeletal developmental defect? Can the authors comment on it?

Response:

Thank you for your thoughtful question. We agree that exploring the basis of the short stature and reduced exercise capacity observed in *Foxk2*^{fl/fl}; *Myod1*-Cre mice can deepen the understanding of the phenotype. The mouse strain used in our study was

designed to conditionally knock out *Foxk2* in muscle stem cells (MuSCs) using *Myod1*-Cre. Among the key myogenic regulatory factors, MYF5 and MYOD are commonly targeted in skeletal muscle development studies (Guo *et al*, 2022; Kim *et al*, 2023). MYF5 is expressed in early myogenesis and contributes to both myogenic and non-myogenic lineages, including chondrocytes and osteoblasts (Guo *et al.*, 2022). In contrast, MYOD plays a more specific role in driving myogenic differentiation, even converting various non-muscle cell types—such as pigment, nerve, fat, and liver cells—into skeletal muscle cells (Weintraub *et al*, 1989). Therefore, we selected the *Myod1*-Cre mouse to ensure specific targeting of myogenic differentiation pathways. Based on this model, the phenotypes of short stature and reduced exercise capacity are primarily attributed to impaired skeletal muscle development, rather than a broader skeletal developmental defect. These findings align with the role of FOXK2 in regulating muscle-specific genes and mitochondrial function in skeletal muscle. We hope this explanation clarifies the underlying cause of the observed phenotypes, and we appreciate your attention to this detail.

2. The authors identified the effect of FOXK2 deficiency on the expression of mitochondrial function-related genes in muscle stem cells and C2C12 cells. Why different sets of genes were chosen? This reviewer would like to see the data currently missing for the other genes.

Response:

Thank you for your thoughtful suggestion. We fully understand the importance of transparency and reproducibility in scientific research. Regarding the analysis of mitochondrial function-related genes in FOXK2 knockout muscle stem cells (MuSCs) and C2C12 cells, we initially selected genes such as *Pgc1a*, *Tfb1m*, *Tfb2m*, *Cox4l* and *Cycs* based on their known roles in mitochondrial biogenesis and function. However, some of these genes did not show significant expression differences, and due to figure layout limitations, we only presented a subset of the data in the original manuscript to maintain focus and clarity. To address your concern, we have now included additional data on the expression levels of these mitochondrial function-related genes in both MuSCs from *Foxk2^{fl/fl}* and *Foxk2^{fl/fl}; Myod1*-Cre mice, as well as in C2C12 cells with various treatments. The updated results are now presented in Figure 4I and Figure EV4C. We hope that these additional data provide a more

comprehensive view of our findings and address your concern. Thank you again for your valuable input.

3. The authors singled out the *Pdk4* and *Cry1* genes based on ATAC-seq without showing their respective peak plots. It would be better to provide these data.

Response:

Thank you very much for your thoughtful comment and for pointing out the need for additional data. We sincerely apologize for not including the ATAC-seq peak plot for *Pdk4* and *Cry1* due to space constraints in the original manuscript. To address your concern, we have now included the ATAC-seq peak plots for these genes in FOXX2 knockout C2C12 cells in Figure 8I. These plots clearly demonstrate that FOXX2 knockout results in increased chromatin accessibility at the binding sites of *Pdk4* and *Cry1*, supporting our previous findings. We appreciate your feedback, as it greatly enhances the accuracy and clarity of our manuscript. Thank you again for helping improve the quality of our work.

4. This study provided convincing data demonstrating that FOXK2 is essential for skeletal muscle development. Meanwhile, the differential expression patterns were observed for FOXK1 and FOXK2 in myogenesis of embryonic mice and myogenic differentiation of C2C12 cells (Figure S6), suggesting that FOXK1 and FOXK2 might be of distinct functions. Is there any concern about the function of FOXK1 in skeletal muscle? The authors need to discuss in appropriate places.

Response:

Thank you for your insightful and thought-provoking comment. FOXK1 and FOXK2 belong to the Forkhead box transcription factor family, and their differential expression patterns during myogenesis suggest that they may have distinct roles in skeletal muscle development. While FOXK1 inhibits myogenic differentiation by repressing myocyte enhancer factor 2 (MEF2) activity (Li *et al*, 2022; Shi *et al*, 2012), its deficiency actually impedes muscle regeneration in mice. In our study, we observed that FOXK2 is expressed earlier than FOXK1 during embryonic skeletal muscle development (Figures S2A and S7A-B), which could indicate that FOXK2 has an earlier role in muscle development. This finding suggests that FOXK2 might be functional before FOXK1, highlighting a potential temporal distinction in their functions. We agree that exploring the differential functions of FOXK1 and FOXK2 in skeletal muscle is a fascinating and important area of research. However, addressing this question in detail would require a separate set of experiments and analysis that are beyond the scope of this current study. To acknowledge this, we have added the following discussion to lines 685-687 of the revised manuscript:

"FOXK2 is expressed at an earlier stage than FOXK1, which suggests that FOXK2

may function earlier in skeletal muscle development."

Thank you again for raising this important point, which we believe could guide future studies on the functional differences between these two transcription factors in muscle biology.

5. The annotation of some figures in the article is not consistent in styles. For example, the annotation of figure 1b "H&E" is outside the diagram and figure 6b "DAPI DRP1" is in the diagram, so it is recommended that the author modify it.

Response:

We sincerely thank you for pointing out the inconsistency in the style of figure annotations. We agree that maintaining a uniform style across all figures is essential for improving the clarity and professionalism of the manuscript. To address this issue, we have carefully revised all figures to ensure consistency in the following aspects:

1. Graphical Elements: We have standardized the placement of descriptors for markers in slice staining by positioning them in the upper left corner of the figures (e.g., the red box in Figure 3F).

2. Presentation of Zoomed-in Areas: Zoomed-in areas are now indicated by a box rather than a text description (e.g., the red box in Figure 1C).

3. Experimental Methods: Instead of repeating experimental methods within the figures, we have moved these descriptions to the figure legends.

We believe these revisions have improved the overall presentation of the figures and enhanced the readability of the manuscript. Thank you again for your valuable feedback, which has helped us improve the quality of our work.

6. Lastly, language should be further polished, which would tremendously benefit the readership. For instance: "pave the way for improved strategies...".

Response:

We sincerely thank you for careful reading of our manuscript and for raising this point regarding grammar. We have reviewed the sentence in question. In line 95-96 of the article, the sentence reads: "Ultimately, these insights pave the way for improved strategies in the prevention and treatment of these conditions." In this context, the word "improved" is used as an adjective, and we believe the current phrasing is grammatically correct and consistent with standard academic writing conventions. However, we appreciate your attention to detail and are happy to revise the sentence if there is any opportunity to improve its clarity or readability. Please let us know if you have a specific revision in mind.

Referee #3

With great interest, I have read the manuscript by Wu et al, which identifies **FOXK2** as a novel pathogenic gene associated with congenital myopathy and ptosis. Through whole exome sequencing (WES) of clinical families and functional studies in animal and cell models, the research reveals the role of **FOXK2** in regulating skeletal muscle development. **FOXK2** is shown to maintain mitochondrial function, which is crucial for the development of muscle stem cells (MuSCs). The study also provides a new perspective on the genetic etiology of congenital ptosis and has broader implications for disorders related to skeletal muscle development.

This is a well written and performed study, that should be published once my comments are suitable addressed:

1. It remains unclear why the authors consider the three variants (**P411A, A479V, R544Q**) from pedigrees 3, 4, and 5 to be pathogenic rather than rare polymorphisms. Firstly, for a congenital, dominant rare disease, the frequency of these variants in the control population seems somewhat high. Secondly, upon screening these three variants (**P411A, A479V, R544Q**) in our local whole-exome sequencing (WES) database, which includes over 2000 unique patients, we found that these variants were present in several individuals without myopathy and ptosis. Thirdly, while in silico predictions suggest that all these variants are likely damaging, further experiment need to be identified to ascertain pathogenicity, such as their impact on protein expression.

Response:

We sincerely thank you for careful reading of our manuscript and apologize for any confusion. We understand the importance of accurately screening variants and addressing concerns about their pathogenicity. Below, we provide a point-by-point response to your comments:

1. Allele Genetic Frequency:

We agree that the allele frequency is an important consideration. For our study, we used an allele frequency threshold of less than 1‰ as the criterion for screening *FOXK2* gene

variants. According to data from ExAC and gnomAD for the East Asian population, the frequencies of the variants in question are as follows: P411A: 0.001 in ExAC_EAS and 0.0008 in gnomAD_EAS, A479V: 0.0003 in ExAC_EAS and 0.0003 in gnomAD_EAS, and R544Q: 0.0001 in ExAC_EAS and 0.0002 in gnomAD_EAS. These frequencies are within the threshold for screening rare variants and justify our consideration of them.

2. Phenotypic Heterogeneity:

We acknowledge that clinical heterogeneity is a major feature of hereditary congenital ptosis. For example, one case study reported monozygotic twins with incomplete concordance for congenital ptosis¹¹. Another study reported a family with autosomal dominant congenital ptosis and 70–90% penetrance¹². Additionally, we are very interested in the phenotype of the patients you have collected. Do they have an etiology related to skeletal muscle development? We would be eager to collaborate with you on this, as it presents an exciting scientific opportunity.

3. Impact on Protein Expression:

In response to your concern regarding the impact of these variants on protein expression, we overexpressed wild-type FO XK2 and the following mutant plasmids in HEK-293T cells: c.1231C>G:p.Pro411Ala (FO XK2-P411A), c.1436C>T:p.A479V (FO XK2-A479V) and c.1631G>A:p.R544Q (FO XK2-R544Q). Western blot analysis revealed that the expression of FO XK2 in FO XK2-P411A and FO XK2-R544Q cells was significantly decreased compared to wild-type cells, suggesting that these mutations lead to FO XK2 protein instability. In contrast, the FO XK2-A479V variant did not show a significant impact on protein expression. As a result, we have removed the A479V variant (pedigree 4) from the manuscript. These updated findings are shown in Figure EV1, and we have revised the manuscript to reflect these results:

"We further assessed these mutations through in vitro western blot experiments. The T55I, R215W, P411A, R544Q and Q550H mutations resulted in reduced FO XK2 protein expression, whereas the A479V mutation did not, leading to its exclusion from further analysis. " (lines 407-409).

We sincerely appreciate your thoughtful feedback, which has not only helped clarify these points but also strengthened the rigor and overall impact of our study.

2. Why does the heterozygous variant p.Arg215Trp result in a reduction of protein levels to an undetectable level (figure 1D)? Additional evidence is required to ascertain the loss of function (LoF), such as the use of point mutation cell models.

Response:

Thank you for your insightful comments regarding the c.643C>T:p.R215W mutation and its impact on protein levels. We understand the importance of clear and reproducible results for our readers and sincerely apologize for any confusion caused. Initially, the undetectable levels of FOXK2 in patient (II-5) samples, as shown in Figure 1D, may have been due to unsuitable exposure times during the western blot imaging process. Additionally, the difficulty of performing western blot experiments on tissues and the long storage times of these tissues may have contributed to the issue. To provide clearer evidence, we have updated this result by using immunohistochemistry to detect FOXK2 expression in eyelid muscle tissue from both control and patient. This new data is now included in Figure 1D.

To further evaluate whether the c.643C>T:p.R215W mutation results in a loss of

function, we conducted the following experiments: We transfected FOXX2 knockout C2C12 cells (FOXX2-KO) with either wild-type FOXX2 plasmid (FOXX2+WT) or the c.643C>T;p.R215W mutation plasmid (FOXX2+R215W). After six days of *in vitro* myogenesis differentiation, myotube formation was assessed *via* MyHC immunofluorescence staining. Additionally, the expression of muscle maturation genes *Myh4* and *Myog* was measured RT-qPCR. The FOXX2+R215W group showed incomplete recovered compared to the FOXX2+WT group (as shown in Figure 5D-E). We also evaluated mitochondrial morphology and function. The FOXX2+R215W group exhibited fragmented mitochondria, excessive mitochondrial fission, and impaired function—including reduced ATP production and elevated ROS levels—compared to the FOXX2+WT group (as shown in Figure 6). These results collectively demonstrate that the c.643C>T;p.R215W mutation leads to a loss of function.

We appreciate your patience and valuable feedback, which have significantly helped us improve the quality and clarity of our study.

3. The pedigree families 3, 4, and 5 exhibit dominant traits. Are there additional members within these families available for genetic segregation analysis?

Response:

We sincerely appreciate your thoughtful suggestion, which has helped refine our

analysis. To address your question, we reached out to our collaborators to inquire about additional members within pedigrees 3, 4, and 5 for genetic segregation analysis. Unfortunately, we were unable to recall members from these families for further analysis. We are grateful for your time and valuable feedback.

4. The detailed phenotypes for each pedigree should be provided, including age at onset, muscle MRI, and muscle histopathological examination. The index patient in pedigrees 3-5 exhibits global developmental delay. What are the phenotypes of their parents? I believe they appear to be milder carriers, as they are typically married and childbearing; Why do phenotypes vary within the same family and across different families (isolated ptosis in pedigree1)?

Response:

Thank you for your thoughtful and insightful comments on the clinical aspects of our study. We appreciate the time and effort you have dedicated to reviewing our manuscript. We recognize the importance of providing comprehensive clinical patient information and apologize for any omissions. Below, we address your points in detail:

1. Age at Onset, Muscle MRI and Histopathological Examination:

Unfortunately, we were unable to obtain age at onset, muscle MRI or histopathological examination for patients in pedigree 2-5, except for H&E staining and immunohistochemistry of eyelid muscle tissue from pedigree 1, as shown in Figure 1B-C.

2. Phenotypes of Parents in Original Pedigrees 3-5:

We reached out to our collaborators to inquire about additional information regarding the parents of the probands in original pedigrees 3-5. Unfortunately, we were unable to recall the parents from these pedigrees for further analysis.

3. Phenotypic Heterogeneity Across Families:

There are several factors that could explain the variation in phenotypes within the same family and across different families:

Mutation Location: Different mutations in the *FOXK2* gene may affect the protein's function differently. Our current analysis of the FOXK2 protein domains is limited to the FHA, FOX, and NLS domains, and mutations outside these regions may have unknown effects (as shown in Figure 1G).

Compensatory Cellular Responses: Some mutations may trigger a compensatory response in cells, upregulating other genes to compensate for the loss of function.

Dominant Negative Effects: Certain mutations could have dominant negative effects, interfering with the normal function of the wild-type allele, leading to clinical heterogeneity.

We have updated the manuscript to reflect these points in lines 673-680:

"However, the functional consequences of other mutations on FOXX2 remain unclear due to the unknown roles of the domains where these mutations occur. Moreover, certain mutations might initiate a compensatory cellular response, upregulating associated genes to counterbalance the loss of function. Additionally, gene products with dominant negative variants could potentially interfere with the normal function of wild-type alleles. These factors may contribute to the clinical heterogeneity observed among different FOXX2 mutations. Further investigation is crucial to clarify the specific impacts of these FOXX2 mutations through functional experiments."

We are grateful for your constructive feedback, which has significantly improved the clarity and academic value of our manuscript. Your suggestions have also provided valuable guidance for further research in this field.

5. Western blot shows that FOXX2 has two bands in Figure 5A, C but only one band in figure 1D, which one is the true target band? The expression level of the R215W mutant plasmid (Figure 5C) is not lower than that of WT, and may even be higher, which is inconsistent with the results of LoF in figure 1D.

Response:

Thank you for your valuable comments and for pointing out these important issues regarding the western blot results. We sincerely apologize for any confusion caused by unclear labeling and inconsistencies. Below, we provide a point-by-point clarification:

1. Target Band of the FOXX2 Antibody:

We consulted technical support from NOVUS Biologicals, the manufacturer of the

FOXK2 antibody (NBP1-87700). This antibody is a polyclonal, and multiple bands are commonly observed in western blot analysis, as also shown on the company's product page. According to their guidance, the 80 kDa band represents the specific FOXK2 target, while the other bands are non-specific.

2. Western Blot in Human Tissues (Figure 1D):

Due to the technical challenges of performing western blot on tissue samples and potential degradation over long-term storage, we may have used inappropriate exposure times, which resulted in an undetectable FOXK2 band in the patient (II-5) group. To address this, we replaced the tissue western blot data in Figure 1D with immunohistochemistry results to more clearly illustrate FOXK2 expression in control vs. patient eyelid muscle tissue.

3. Expression Level of R215W Mutant in Figure 5C:

We agree that the FOXK2 band in the R215W mutant group appeared similar to or even higher than the wild-type in the original blot, which created inconsistency with our LoF interpretation. Upon review, we found that the blot in Figure 5C contained non-specific thick bands, likely due to partial degradation of the protein samples, leading to inaccurate signal interpretation. To resolve this, we repeated the experiment using freshly prepared cell lysates to avoid degradation. The updated results show that FOXK2 expression in KO+R215W cells is indeed reduced compared to KO+WT,

consistent with the hypothesis that the R215W mutation impairs FOXX2 protein stability. The corrected data has been included in the revised Figure 5C.

We deeply appreciate your careful review and helpful critique, which have allowed us to identify and correct these issues in a timely manner. We are committed to presenting accurate and high-quality data and believe the revised figures and explanations now more clearly support our conclusions.

6. It is recommended to provide quantification of mitochondrial morphology through confocal imaging of labeled mitochondria. Additionally, Western blot analysis of key proteins involved in mitochondrial dynamics should also be supplied.

Response:

Thank you for your insightful suggestion regarding the need for additional mitochondrial data. We appreciate your feedback and have addressed this by including the following:

1. Confocal Imaging of Labeled Mitochondria:

We have added confocal microscopy imaging to evaluate mitochondrial morphology (mito-tracker staining). The results show that FOXX2-KO cells exhibit an increased number of smaller mitochondria, consistent with the observed phenotype of excessive mitochondrial fission compared to control cells.

2. Western Blot Analysis of Mitochondrial Dynamics Proteins:

We have also performed western blot analysis of key proteins involved in mitochondrial dynamics, specifically OPA1 and DRP1. The data indicate that DRP1 is upregulated, while OPA1 is downregulated in FOXX2-KO cells, further supporting our conclusion

that loss of FOXK2 leads to an imbalance in mitochondrial dynamics. These new data have been added to Figure EV4A-B. We believe these additions enhance the clarity and completeness of our findings.

We sincerely appreciate your helpful comments, which have contributed to improving the manuscript.

References

Guo R, You X, Meng K, Sha R, Wang Z, Yuan N, Peng Q, Li Z, Xie Z, Chen R *et al* (2022) Single-Cell RNA Sequencing Reveals Heterogeneity of Myf5-Derived Cells and Altered Myogenic Fate in the Absence of SRSF2. *Adv Sci (Weinh)* 9: e2105775

Guo YL, Chakraborty S, Rajan SS, Wang R, Huang F (2010) Effects of oxidative stress on mouse embryonic stem cell proliferation, apoptosis, senescence, and self-renewal. *Stem Cells Dev* 19: 1321-1331

Kim KH, Oprescu SN, Snyder MM, Kim A, Jia Z, Yue F, Kuang S (2023) PRMT5 mediates FoxO1 methylation and subcellular localization to regulate lipophagy in myogenic progenitors. *Cell Rep* 42: 113329

Li C, Shen H, Liu M, Li S, Luo Y (2022) Natural antisense RNA Foxk1-AS promotes myogenic differentiation by inhibiting Foxk1 activity. *Cell Commun Signal* 20: 77

Liu X, Tang N, Liu Y, Fu J, Zhao Y, Wang H, Wang H, Hu Z (2023) FOXK2 regulates PFKFB3 in promoting glycolysis and tumorigenesis in multiple myeloma. *Leuk Res* 132: 107343

Shi X, Wallis AM, Gerard RD, Voelker KA, Grange RW, DePinho RA, Garry MG, Garry DJ (2012) Foxk1 promotes cell proliferation and represses myogenic differentiation by regulating Foxo4 and Mef2. *J Cell Sci* 125: 5329-5337

Sukonina V, Ma H, Zhang W, Bartesaghi S, Subhash S, Heglind M, Foyn H, Betz MJ, Nilsson D, Lidell ME *et al* (2019) FOXK1 and FOXK2 regulate aerobic glycolysis. *Nature* 566: 279-283

Turner KL, Gheres KW, Drew PJ (2023) Relating Pupil Diameter and Blinking to Cortical Activity and Hemodynamics across Arousal States. *J Neurosci* 43: 949-964

Weintraub H, Tapscott SJ, Davis RL, Thayer MJ, Adam MA, Lassar AB, Miller AD (1989) Activation of muscle-specific genes in pigment, nerve, fat, liver, and fibroblast cell lines by forced expression of MyoD. *Proc Natl Acad Sci U S A* 86: 5434-5438

15th Apr 2025

Dear Prof. Ma,

Thank you for the submission of your manuscript to EMBO Molecular Medicine. I am pleased to inform you that we will be able to accept your manuscript pending the following final amendments:

- 1) Authors: We note that you currently have 3 first authors and together with you, a total of 4 co-corresponding authors. Is that correct? Do you confirm equal contribution of these authors, able to take full responsibility for the paper and its content? While there is no limit per se to the number of first and co-corresponding authors, 3 first and 4 co-corresponding authors is rare, and may not reflect as intended to the community.
- 2) Figures: Please rename EV figure files to Figure EV1, etc.
- 3) In the main manuscript file, please do the following:
 - Please address all comments suggested by our data editors listed below:
 - o Data availability statement:
 1. Please note that the specific URLs for S-BSST1949, GSE291642 datasets are not provided.
 - o Figure legends:
 1. Please note that the exact p values are not provided in the legends of figures 1C, 2B, E, H, I, J, K; 3B, C, D, E, F, G; 4B, C, D, E, F, G, H, I; 5D, E; 6B, E, F, G, H, I; 7B, C, D, F, G, J; EV3 A, B; EV4 A; EV5 C, D, E, H; S2 F-H; S3 A, B, C.
 2. Please indicate the statistical test used for data analysis in the legends of figures 1C, 2B, E, H, I, J, K; 3B, C, D, E, F, G; 4B, C, D, E, F, G, H, I; 5D, E; 6B, E, F, G, H, I; 7B, C, D, F, G; 8C, F, H, J; EV3 A, B; EV4A, C; EV5 A, C, D, E, H; S2 F-H; S3 A, B, C".
 3. Please note that information related to n is missing in the legends of figures 5B, 8H, EV5 C.
 4. Please note that the error bars are not defined in the legend of figure S4 A.
 5. Please note that the scale bar needs to be defined for figures 4F.
 - Limit keywords to max. 5.
 - The manuscript sections should be in the following order: Title page - Abstract & Keywords - Introduction - Results - Discussion - Methods - Data Availability - Acknowledgments - Disclosure Statement & Competing Interests - References - Figure Legends - (Main Tables with legends if applicable) - Expanded View Figure Legends.
 - Remove all figures.
 - In Methods, add statistical paragraph that should reflect all information that you have filled in the Authors Checklist, especially regarding randomization, blinding, replication etc.
 - Please check our guide to authors regarding publication of identifiable formats of human subjects (video, recording, photograph, image). <https://www.embopress.org/page/journal/17574684/authorguide#humansubjects>
 - Indicate in legends exact n and exact p values, not a range, along with the statistical test used. To keep the figures "clear" some authors found providing an Appendix table Sx with all exact p-values preferable. You are welcome to do this if you want to.
 - Rename "Conflict of interest" to "Disclosure Statement & Competing Interests". We updated our journal's competing interests policy in January 2022 and request authors to consider both actual and perceived competing interests. Please review the policy <https://www.embopress.org/competing-interests> and update your competing interests if necessary.
 - Author contributions: Please remove it from the manuscript and specify author contributions in our submission system. CRediT has replaced the traditional author contributions section because it offers a systematic machine-readable author contributions format that allows for more effective research assessment. You are encouraged to use the free text boxes beneath each contributing author's name to add specific details on the author's contribution. More information is available in our guide to authors:
<https://www.embopress.org/page/journal/17574684/authorguide#authorshippinguidelines>
 - Data availability: Please use the following format to report the accession number of your data:

[data type]: [full name of the resource] [accession number/identifier] ([doi or URL or identifiers.org/DATABASE:ACCESSION])

Please check "Author Guidelines" for more information.

<https://www.embopress.org/page/journal/17574684/authorguide#availabilityofpublishedmaterial>

4) Tables: Please rename Tables EV3, EV4 EV5, EV6 to Dataset EV1-EV4 with their legends in the separate tab and update their callouts in the main text. Tab EV1 and EV2 with their legends should be placed in the Appendix after appendix figures, renamed to Appendix Table S1 and S2 and their callouts in the main text updated. Table EV7 should be renamed to Table EV1 with its legend in the same sheet with the table and its callout updated in the main text.

5) Appendix: Please add page numbers also in the table of content.

6) The Paper Explained: Please add it to the main manuscript file.

7) Synopsis:

- Synopsis image: Please provide a visual abstract as a schematic with no or only few figure panels (with appropriate dimensions, current ones are too small) and little text as possible. Please upload it as a high-resolution jpeg file 550 pixels wide x 200-600 pixels high to illustrate your article.

8) As part of the EMBO Publications transparent editorial process initiative (see our Editorial at <http://embomolmed.embopress.org/content/2/9/329>), EMBO Molecular Medicine will publish online a Review Process File (RPF) to accompany accepted manuscripts. This file will be published in conjunction with your paper and will include the anonymous referee reports, your point-by-point response and all pertinent correspondence relating to the manuscript. Let us know whether you agree with the publication of the RPF and as here, if you want to remove or not any figures from it prior to publication. Please note that the Authors checklist will be published at the end of the RPF.

9) Please provide a point-by-point letter INCLUDING my comments as well as the reviewer's reports and your detailed responses (as Word file).

I look forward to reading a new revised version of your manuscript as soon as possible.

Yours sincerely,

Zeljko Durdevic

Zeljko Durdevic
Senior Editor
EMBO Molecular Medicine

*** Instructions to submit your revised manuscript ***

1) a .docx formatted version of the manuscript text (including Figure legends and tables)

2) Separate figure files*

3) supplemental information as Expanded View and/or Appendix. Please carefully check the authors guidelines for formatting Expanded view and Appendix figures and tables at <https://www.embopress.org/page/journal/17574684/authorguide#expandedview>

4) a letter INCLUDING the reviewer's reports and your detailed responses to their comments (as Word file).

5) The paper explained: EMBO Molecular Medicine articles are accompanied by a summary of the articles to emphasize the major findings in the paper and their medical implications for the non-specialist reader. Please provide a draft summary of your article highlighting

6) Author contributions: the contribution of every author must be detailed in a separate section.

7) EMBO Molecular Medicine now requires a complete author checklist (<https://www.embopress.org/page/journal/17574684/authorguide>) to be submitted with all revised manuscripts. Please use the checklist as guideline for the sort of information we need WITHIN the manuscript. The checklist should only be filled with page numbers where the information can be found. This is particularly important for animal reporting, antibody dilutions (missing) and exact values and n that should be indicated instead of a range.

8) Every published paper now includes a 'Synopsis' to further enhance discoverability. Synopses are displayed on the journal webpage and are freely accessible to all readers. They include a short stand first (maximum of 300 characters, including space) as well as 2-5 one sentence bullet points that summarise the paper. Please write the bullet points to summarise the key NEW findings. They should be designed to be complementary to the abstract - i.e. not repeat the same text. We encourage inclusion of key acronyms and quantitative information (maximum of 30 words / bullet point). Please use the passive voice. Please attach these in a separate file or send them by email, we will incorporate them accordingly.

You are also welcome to suggest a striking image or visual abstract to illustrate your article. If you do please provide a jpeg file 550 px-wide x 300-600px high.

9) A Conflict of Interest statement should be provided in the main text

10) Please note that we now mandate that all corresponding authors list an ORCID digital identifier. This takes <90 seconds to complete. We encourage all authors to supply an ORCID identifier, which will be linked to their name for unambiguous name identification.

Currently, our records indicate that the ORCID for your account is 0000-0001-9074-8570.

Link Not Available

11) Include a Reagents and Tools Table as part of the Methods section, which can be downloaded from our author guidelines (<https://www.embopress.org/page/journal/17574684/authorguide#structuredmethods>)

Photos 400-800 DPI

*Additional important information regarding figures and illustrations can be found at <https://bit.ly/EMBOPressFigurePreparationGuideline>. See also figure legend preparation guidelines: <https://www.embopress.org/page/journal/17574684/authorguide#figureformat>

***** Reviewer's comments *****

Referee #1 (Comments on Novelty/Model System for Author):

The authors did a comprehensive evaluation of this conditional Foxk2 MyoD1 Cre KO mice. The authors also incorporated human patient data. I don't think any alternative models could be used.

Referee #1 (Remarks for Author):

The authors have provided a comprehensive response to all of my previous comments as well as the other reviewers. The manuscript has extensive analysis of the ROS and glycolytic defects in addition to the myotube/myogenic differentiation defects. The manuscript was already an excellent manuscript and has been enhanced by the additional data and experimentation. I have no further edits and believe the manuscript is highly suitable for EMBO Molecular Medicine.

Referee #2 (Remarks for Author):

Is suitable for publication.

The authors addressed the remaining editorial issues.

29th Apr 2025

Dear Prof. Ma,

Please find enclosed the final reports on your manuscript. We are pleased to inform you that your manuscript is accepted for publication and is now being sent to our publisher to be included in the next available issue of EMBO Molecular Medicine.

Yours sincerely,
